# Model Selection in Contextual Stochastic Bandit Problems

**Aldo Pacchiano**[*]
UC Berkeley

**My Phan**[*]
University of Massachusetts

**Yasin Abbasi-Yadkori**
DeepMind

**Anup Rao**
Adobe

**Julian Zimmert**
Google Research

**Tor Lattimore**
DeepMind

**Csaba Szepesvári**
DeepMind and University of Alberta

## Abstract

We study bandit model selection in stochastic environments. Our approach relies on a master algorithm that selects between candidate base algorithms. We develop a master-base algorithm abstraction that can work with general classes of base algorithms and different type of adversarial master algorithms. Our methods rely on a novel and generic smoothing transformation for bandit algorithms that permits us to obtain optimal $O(\sqrt{T})$ model selection guarantees for stochastic contextual bandit problems as long as the optimal base algorithm satisfies a high probability regret guarantee. We show through a lower bound that even when one of the base algorithms has $O(\log T)$ regret, in general it is impossible to get better than $\Omega(\sqrt{T})$ regret in model selection, even asymptotically. Using our techniques, we address model selection in a variety of problems such as misspecified linear contextual bandits [13], linear bandit with unknown dimension [8] and reinforcement learning with unknown feature maps. Our algorithm requires the knowledge of the optimal base regret to adjust the master learning rate. We show that without such prior knowledge any master can suffer a regret larger than the optimal base regret.

## 1 Introduction

In a bandit model selection problem, given a set of base algorithms, the learner aims to adapt in an online fashion to the best base that is the most suitable for the current environment. Maillard and Munos [15] are perhaps the first to address the bandit model-selection problem, with a variant of EXP4 master algorithm that works with UCB or EXP3 base algorithms. These results are improved by Agarwal et al. [2]. Agarwal et al. [2] combine the base algorithms using an online mirror descent master (CORRAL) that sends importance weighted rewards to the base algorithms, thus requiring each base algorithm to be individually modified to be compatible with the master. For example, to use UCB as a base, we would need to manually re-derive UCB's confidence interval and modify its regret analysis to be compatible with importance weighted feedback. Instead, we introduce a generic smoothing wrapper method that can be applied to base algorithms without substantial modification.

There are works on model selection in settings such as in linear bandits with unknown dimension or structure [8, 6]. Apart from strong assumptions, those works are limited to a specific model-selection problem. A general and efficient method to combine multiple base algorithms is missing.

**Contributions.** We focus on bandit model-selection in stochastic environments. Our contributions are as follows:

---

[*]Equal contribution.

- We introduce a general "smoothing" wrapper so that any contextual base algorithm can be compatible with the CORRAL [2] and EXP3.P masters [5]. This is more general than the approach of Agarwal et al. [2] where each base algorithm needs to be individually modified to satisfy certain stability condition. Our modification of the CORRAL algorithm has another important difference: instead of importance weighted feedback, the original rewards are sent to the base algorithms. The resulting model selection strategy can be readily used with almost any base algorithm developed for stochastic environments. When the optimal base regret is known, the CORRAL master achieves optimal regret guarantees. Under certain conditions when the optimal base regret is unknown EXP3.P can achieve better performance.

- We demonstrate the generality and effectiveness of our method by showing how it seamlessly improves existing results or addresses open questions in a variety of problems. We show applications in adapting to the misspecification level in contextual linear bandits [13], adapting to the unknown dimension in nested linear bandit classes [8], tuning the data-dependent exploration rate of bandit algorithms, and choosing feature maps in reinforcement learning. Moreover, our master algorithm can simultaneously perform different types of model selection. For example, we show how to choose both the unknown dimension and the unknown mis-specification error at the same time. This is in contrast to algorithms that specialize in a specific type of model selections such as detecting the unknown dimension [8].

- In the stochastic domain, an important question is whether a model selection procedure can inherit the $O(\log T)$ regret of a fast stochastic base algorithm. We show a lower bound for the model selection problem that scales as $\Omega(\sqrt{T})$, which implies that our result is minimax optimal. Our master algorithm requires knowledge of the best base's regret to achieve the same regret. We show that this condition is unavoidable in general: there are problems where regret of the best base scales as $O(T^x)$ for an unknown $x$, and the regret of any master algorithm scales as $\Omega(T^y)$ for $y > x$.

## 2 Problem statement

Let $\delta_a$ denotes the delta distribution at $a$. For an integer $n$, we use $[n]$ to denote the set $\{1, 2, \ldots, n\}$. We consider contextual stochastic bandit problems: Let $\mathcal{A} \subseteq \mathbb{R}^d$ be a set of actions. Let $S$ be the set of all subsets of $\mathcal{A}$ and let $\mathcal{D}_S$ be a distribution over $S$. At time $t$, the learner observes an action set $\mathcal{A}_t$ (which could be infinite) sampled from $\mathcal{D}_S$. The learner chooses policy $\pi_t$, which takes an action set $\mathcal{X} \in S$ as an input and outputs a distribution over $\mathcal{X}$. The learner selects action $a_t \sim \pi_t(\mathcal{A}_t)$ and receives a reward $r_t$ such that $r_t = f(\mathcal{A}_t, \delta_{a_t}) + \xi_t$ where $\xi_t$ is 1-subGaussian random noise and $f(\mathcal{X}, \pi)$ denotes the expected reward of applying policy $\pi$ on action set $\mathcal{X}$. The fixed action case $\mathcal{A}_t = \mathcal{A}$ and the linear contextual bandit problem with IID contexts are special cases of this setting. For linear contextual bandits, there are $k$ actions and a linearly parameterized policy $\pi$ maps from the space of $d \times k$ matrices to $[k]$: in round $t$ and given context $x_t \in \mathbb{R}^{d \times k}$, $\pi_\theta(x_t) = \operatorname{argmax}_{i \in [k]} x_{t,i}^\top \theta$, where $x_{t,i}$ denotes the $i$−th column of $x_t$. Letting $i_t = \pi_\theta(x_t)$, the reward is given by $r_t = x_{t,i_t}^\top \theta^* + \xi_t$ where $\theta^* \in \mathbb{R}^d$ is an unknown parameter vector.

We are interested in designing an algorithm with small regret, defined as

$$R(T) = \max_{\pi^* \in \Pi} \mathbf{E} \left[ \sum_{t=1}^{T} f(\mathcal{A}_t, \pi^*) - \sum_{t=1}^{T} f(\mathcal{A}_t, \pi_t) \right]. \tag{1}$$

We assume there are $M$ candidate *base* algorithms and a master algorithm $\mathcal{M}$ that selects one of the base algorithms in each round. Let $\{p_1^i, \ldots, p_T^i\}$ be the (random) probabilities that $\mathcal{M}$ chooses base $i$ during the game and let $\underline{p}_i = \min_t p_t^i$. If base $\mathcal{B}_i$ satisfies a high probability regret bound $U_i(T, \delta)$ when played in an environment $\mathcal{E}$, we call $\mathcal{E}$ a $U_i$−compatible environment for $\mathcal{B}_i$. We do not require $\mathcal{E}$ to be $U_i$ compatible w.r.t all input base algorithms. We want to design a master algorithm that satisfies regret $R(T) \leq O(U_{i_*}(T, \delta))$; We are interested in competing with the best performing compatible base $i_*$. For the rest of the paper we use $i$ to denote the optimal base $i_*$.

## 3 Main results

We consider the following abstraction in Algorithms 1 and 2. Base algorithms only play their chosen action, receive rewards and update their policy when selected by the master. Base algorithms keep a

counter $s$, keeping track of the number of times they have been invoked. For any base algorithm $\mathcal{B}_j$, $\pi_{s,j}$ is the policy $\mathcal{B}_j$ uses at state $s$. Let $s_{t,j}$ denote the state of base $j$ at time $t$. If $t_1 < t_2$ are two consecutive times when base $j$ is chosen by the master, then the base has policy $\pi_{s_{t_1,j},j}$ at time $t_1$ and policy $\pi_{s_{t_2,j},j}$ at times $t_1 + 1, \ldots, t_2$ where $s_{t_2,j} = s_{t_1,j} + 1$

---

**Algorithm 1** Master Algorithm

**Input:** Base Algorithms $\{\mathcal{B}_j\}_{j=1}^M$
**for** $t = 1, \cdots, T$ **do**
    Play base $j_t$.
    Receive feedback $r_t = r_{t,j_t}$ from $\mathcal{B}_{j_t}$
    Update itself using $r_t$
**end for**

---

**Algorithm 2** Base Algorithm $\mathcal{B}_j$

Initialize state counter $s = 1$
**for** $t = 1, \cdots, T$ **do**
    Receive action set $\mathcal{A}_t \sim \mathcal{D}_S$
    Choose action $a_{t,j} \sim \pi_{s,j}(\mathcal{A}_t)$
    **if** selected by master **then**
        Play action $a_{t,j}$
        Receive feedback $r_{t,j} = f(\mathcal{A}_t, \delta_{a_{t,j}}) + \xi_t$
        Send $r_{t,j}$ to the master
        Compute $\pi_{s+1,j}$ using $r_{t,j}$
        $s \leftarrow s + 1$
    **end if**
**end for**

---

To analyze the regret of the master w.r.t. the optimal base $\mathcal{B}_i$, we add and subtract terms $\{f(\mathcal{A}_t, \pi_{s_{t,i},i})\}_{t=1}^T$ and use a regret decomposition similar to the one used by Agarwal et al. [2]:

$$
R(T) = \mathbf{E}\left[\sum_{t=1}^T f(\mathcal{A}_t, \pi^*) - f(\mathcal{A}_t, \pi_t)\right]
$$

$$
= \mathbf{E}\underbrace{\left[\sum_{t=1}^T f(\mathcal{A}_t, \pi_{s_{t,i},i}) - f(\mathcal{A}_t, \pi_t)\right]}_{\text{I}} + \mathbf{E}\underbrace{\left[\sum_{t=1}^T f(\mathcal{A}_t, \pi^*) - f(\mathcal{A}_t, \pi_{s_{t,i},i})\right]}_{\text{II}} \quad (2)
$$

Term I is the regret of the master with respect to the optimal base, and term II is the regret of the optimal base with respect to the optimal policy $\pi^*$. Analysis of term I is largely based on existing analysis of CORRAL and EXP3.P. To bound term II, we provide a smoothing transformation (Section 5, Algorithm 3) that converts any base algorithm with high probability bound $U(T, \delta)$ to one with high probability instantaneous regret bound $u_t = \frac{U(t,\delta)}{t}$ at time $t$, which is decreasing if $U(T, \delta)$ is concave. Since $u_t$ is decreasing, term II is the largest when base $i$ is selected the least often ($p_t^i = \underline{p}_i \ \forall t$). In this case base $i$ will be played roughly $T\underline{p}_i$ times, and will repeat its decisions in intervals of length $\frac{1}{\underline{p}_i}$, resulting in the following bound:

**Lemma 3.1** (informal). *If regret of the optimal base is bounded by $U_*(T, \delta)$ with probability at least $1 - \delta$ when it runs alone, then we have that $\mathbf{E}\left[\text{II}\right] \leq O\left(\mathbf{E}\left[\frac{1}{\underline{p}_i} U_*(T\underline{p}_i, \delta) \log T\right] + \delta T(\log T + 1)\right).$*

We demonstrate the effectiveness of our smooth transformation by deriving regret bounds with two master algorithms: CORRAL (introduced by Agarwal et al. [2] and reproduced in Appendix B) and EXP3.P (Theorem 3.3 in [5]), a simple algorithm that ensures each base is picked with at least a (horizon dependent) constant probability $p$.

**Theorem 3.2** (informal version of Theorem 5.3). *If $U_*(T, \delta) = O(c(\delta) T^\alpha)$ for some function $c : \mathbb{R} \to \mathbb{R}$ and constant $\alpha \in [1/2, 1)$, the regrets of EXP3.P and CORRAL are:*

|  | Known $\alpha$ and $c(\delta)$ | Known $\alpha$, Unknown $c(\delta)$ |
|---|---|---|
| *EXP3.P* | $\tilde{O}\left(T^{\frac{1}{2-\alpha}} c(\delta)^{\frac{1}{2-\alpha}}\right)$ | $\tilde{O}\left(T^{\frac{1}{2-\alpha}} c(\delta)\right)$ |
| *CORRAL* | $\tilde{O}\left(T^\alpha c(\delta)\right)$ | $\tilde{O}\left(T^\alpha c(\delta)^{\frac{1}{\alpha}}\right)$ |

CORRAL has optimal regret when $\alpha$ and $c(\delta)$ are known. However when $c(\delta)$ is unknown and $c(\delta) > T^{\frac{(1-\alpha)\alpha}{2-\alpha}}$ (which is $T^{1/6}$ if $\alpha = 1/2$ or $\alpha = 1/3$), then EXP3.P has better regret because $\tilde{O}\left(T^{\frac{1}{2-\alpha}} c(\delta)\right) < \tilde{O}\left(T^\alpha c(\delta)^{\frac{1}{\alpha}}\right)$.

**Lower bounds.** Theorem 6.2 shows that if the regret of the best base is $O(T^x)$, in the worst case a master algorithm that does not know $x$ can have regret $\Omega(T^y)$ with $y > x$. Theorem 6.1 shows that in general it is impossible for any master algorithm to achieve a regret better than $\Omega(\sqrt{T})$ even when the best base has regret $O(\log(T))$. When the regret of the best base is $O(\sqrt{T})$, CORRAL with our smoothing achieves the optimal $O(\sqrt{T})$ regret.

The detailed description of the smoothing procedure and the analysis are postponed to Section 5. First, we show some applications of our main result in the next section. All algorithms presented in the next section are compatible with the smoothing procedure, and all regret bounds are direct applications of Theorem 3.2.

## 4 Applications

### 4.1 Misspecified Contextual Linear Bandit

We consider the misspecified linear bandit problem. The learner selects an action $a_t \in \mathcal{A}_t$ and receives a reward $r_t$ such that $|\mathbf{E}[r_t] - a_t^\top \theta| \leq \epsilon_*$ where $\theta \in \mathbb{R}^d$ is an unknown parameter vector and $\epsilon_*$ is the misspecification error. For this problem, [20] and [13] present variants of LinUCB that achieve a high probability $\tilde{O}(d\sqrt{T} + \epsilon_*\sqrt{dT})$ regret bound. Both algorithms require knowledge of $\epsilon_*$, but [13] show a regret bound of the same order without the knowledge of $\epsilon_*$ for the version of the problem with a fixed action set $\mathcal{A}_t = \mathcal{A}$. Their method relies on G-optimal design, which does not work for contextual settings. It is an open question whether it is possible to achieve the above regret without knowing $\epsilon_*$ for problems with changing action sets.

In this section, we show a $\tilde{O}(d\sqrt{T} + \epsilon_*\sqrt{dT})$ regret bound for linear bandit problems with changing action sets without knowing $\epsilon_*$. For problems with fixed action sets, we show an improved regret that matches the lower bound of [12].

Given a constant $E$ so that $|\epsilon_*| \leq E$, we divide the interval $[1, E]$ into an exponential grid $\mathcal{G} = [1, 2, 2^2, ..., 2^{\log(E)}]$. We use $\log(E)$ modified LinUCB bases, from either Zanette et al. [20] or Lattimore et al. [13], with each base algorithm instantiated with a value of $\epsilon$ in the grid.

**Theorem 4.1.** *For the misspecified linear bandit problem described above, the regret of CORRAL with learning rate $\eta = \frac{1}{\sqrt{T}d}$ applied with modified LinUCB base algorithms with $\epsilon \in \mathcal{G}$, is upper bounded by $\tilde{O}(d\sqrt{T} + \epsilon_*\sqrt{dT})$. When the size $k$ action sets are fixed and $\sqrt{k} > d$, the regret of CORRAL with $\eta = \frac{1}{\sqrt{T}d}$ applied with one UCB base and one G-optimal base algorithm [13] is upper bounded by $\tilde{O}\left(\min\left(\frac{k}{d}\sqrt{T}, d\sqrt{T} + \epsilon_*\sqrt{dT}\right)\right)$.*

This result matches the following lower bound that shows that it is impossible to achieve $\tilde{O}(\min(\sqrt{kT}, d\sqrt{T} + \epsilon_*\sqrt{dT}))$ regret:

**Lemma 4.2** (Implied by Theorem 24.4 in [12]). *Let $R_\nu(T)$ denote the cumulative regret at time $T$ on environment $\nu$. For any algorithm, there exists a 1-dimensional linear bandit environment $\nu_1$ and a $k$-armed bandit environment $\nu_2$ such that $R_{\nu_1}(T) \cdot R_{\nu_2}(T) \geq T(k-1)e^{-2}$.*

**Experiment (Figure 1).** Let $d = 2$. Consider a contextual bandit problem with $k = 50$ arms, where each arm $j$ has an associated vector $a_j \in \mathbb{R}^d$ sampled uniformly at random from $[0, 1]^d$. We consider two cases: (1) For a $\theta \in \mathbb{R}^d$ sampled uniformly at random from $[0, 1]^d$, reward of arm $j$ at time $t$ is $a_j^\top \theta + \eta_t$, where $\eta_t \sim N(0, 1)$, and (2) There are $k$ parameters $\mu_j$ for $j \in [k]$ all sampled uniformly at random from $[0, 10]$, so that the reward of arm $j$ at time $t$ is sampled from $N(\mu_j, 1)$. We use CORRAL with learning rate $\eta = \frac{2}{\sqrt{T}d}$ and UCB and LinUCB as base algorithm. In case (1) LinUCB performs better while in case (2) UCB performs better. Each experiment is repeated 500 times.

### 4.2 Contextual Bandits with Unknown Dimension

**Linear Contextual Bandit.** We consider the contextual linear bandit problem studied by Foster et al. [8]. In this problem, each action $a \in \mathcal{A}_t$ is a $D$-dimensional feature map, but only the first $d_*$ elements of the parameter vector are nonzero. Here, $d_*$ is unknown and possibly much smaller than $D$. [8] consider the special case when the number of actions $k$ is finite and require a lower bound

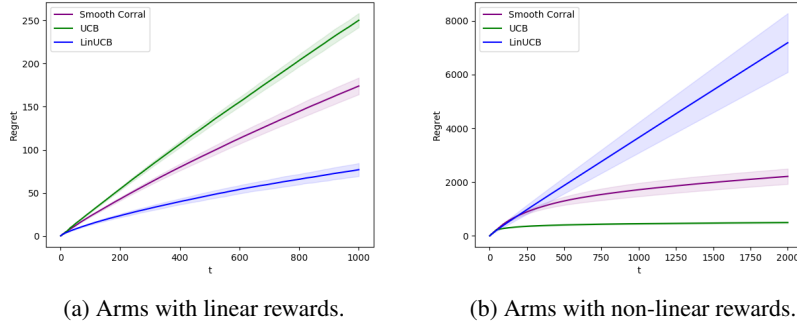

(a) Arms with linear rewards.　　　　(b) Arms with non-linear rewards.

Figure 1: CORRAL with UCB and LinUCB bases. Shaded regions denote the standard deviations.

on the average eigenvalues of the co-variance matrices of all actions. We provide the first sublinear regret for this problem when the action set is infinite. Further, we have no eigenvalue assumptions and our regret does not scale with the number of actions $k$.

We use LinUCB with each value of $d \in [1, 2, 2^2, ..., 2^{\log(D)}]$ as a base algorithm for CORRAL and EXP3.P. We also consider the case when both the optimal dimension $d_*$ and the misspecification $\epsilon_*$ are unknown: we use $M = \log(E) \cdot \log(D)$ modified LinUCB bases (see Section 4.1) for each value of $(\epsilon_*, d_*)$ in the grid $[1, 2, 2^2, ..., 2^{\log(E)}] \times [1, 2, 2^2, ..., 2^{\log(D)}]$. We obtain the regret bounds summarized in the following table:

| | Linear contextual bandit | | Misspecified linear contextual bandit |
|---|---|---|---|
| | Unknown $d_*$ | | Unknown $d_*$ and $\epsilon_*$ |
| | Finite action sets | Infinite action sets | |
| Foster et al. [8] | $\tilde{O}(T^{2/3}k^{1/3}d_*^{1/3})$ or $\tilde{O}(k^{1/4}T^{3/4} + \sqrt{kTd_*})$ | N/A | N/A |
| EXP3.P | $\tilde{O}(d_*^{\frac{1}{2}}T^{\frac{2}{3}})$ | $\tilde{O}(d_*T^{\frac{2}{3}})$ | $\tilde{O}(T^{\frac{2}{3}}d_* + \epsilon_*\sqrt{d}T)$ |
| CORRAL | $\tilde{O}\left(d_*\sqrt{T}\right)$ | $\tilde{O}\left(d_*^2\sqrt{T}\right)$ | $\tilde{O}\left(\sqrt{T}d_*^2 + \epsilon_*\sqrt{d}T\right)$ |

With our approach, it is possible to combine different types of master and base algorithms, which provides much more flexibility compared to approaches specializing in a specific type of model selection. To the best of our knowledge, this is the first result that provides these types of guarantees.

**Nonparametric Contextual Bandit.** [9] consider non-parametric stochastic contextual bandits. At time $t$ and given a context $x_t \in \mathbb{R}^N$, the learner selects arm $a_t \in [k]$ and observes the reward $f_{a_t}(x_t) + \xi_t$, where $\xi_t$ is a 1-sub-Gaussian random variable and $f_j$ denotes the mean reward function of arm $j$. It is assumed that the contexts arrive in an IID fashion. [9] obtain a $\tilde{O}\left(T^{\frac{1+N}{2+N}}\right)$ regret for this problem. Similar to Foster et al. [8], we assume that only the first $n_*$ context features are relevant for an unknown $n_* < N$. It is important to find $n_*$ because $T^{\frac{1+n_*}{2+n_*}} \ll T^{\frac{1+N}{2+N}}$. We have a model selection strategy that adapts to this unknown quantity: for each value of $n$ in the grid $[b^0, b^1, b^2, ..., b^{\log_b(N)}]$ for some $b > 1$, we use the algorithm of Guan and Jiang [9] as a base, and perform model selection with CORRAL and EXP3.P with these base algorithms.

| | Foster et al. [8] | EXP3.P | CORRAL |
|---|---|---|---|
| Nonparametric contextual bandit Unknown $n_*$ | N/A | $\tilde{O}\left(T^{\frac{1+bn_*}{2+bn_*} + \frac{1}{3(2+bn_*)}}\right)$ | $\tilde{O}\left(T^{\frac{1+2bn_*}{2+2bn_*}}\right)$ |

### 4.3 Tuning the exploration rate of $\epsilon$-greedy

For a given positive constant $c$, the $\epsilon$-greedy algorithm pulls the arm with the largest empirical average reward with probability $1 - c/t$, and otherwise pulls an arm uniformly at random. Let $\epsilon_t = c/t$. It can be shown that the optimal value for $\epsilon_t$ is $\min\{1, \frac{5k}{\Delta_*^2 t}\}$ where $\Delta_*$ is the smallest gap between the optimal arm and the sub-optimal arms [12]. With this exploration rate, the regret scales as $\widetilde{O}(\sqrt{T})$ for $k = 2$. We would like to find the optimal value of $c$ without the knowledge of $\Delta_*$. We obtain

such result by applying CORRAL to a set of $\epsilon$-greedy base algorithms each instantiated with a $c$ in $[1, 2, 2^2, ..., 2^{\log(kT)}]$.

**Theorem 4.3.** *The regret of CORRAL using $\epsilon$-greedy base algorithms defined on the grid is bounded by $\tilde{O}(T^{1/2})$ when $k = 2$.*

**Experiment (Figure 2).** Let there be two Bernoulli arms with means $0.5$ and $0.45$. We use 18 $\epsilon$-greedy base algorithms differing in their choice of $c$ in the exploration rate $\epsilon_t = c/t$. We take $T = 50,000$, $\eta = 20/\sqrt{T}$ and $\epsilon$'s to lie on a geometric grid in $[1, 2T]$. Each experiments is repeated 50 times.

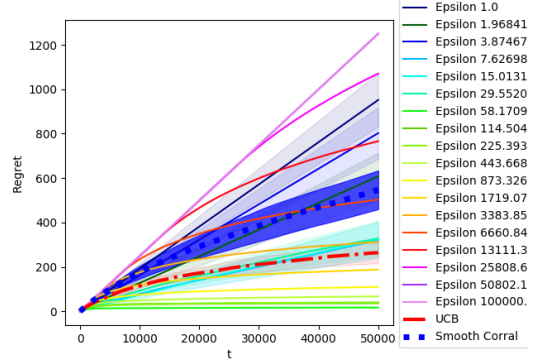

### 4.4 Reinforcement Learning

We consider the case of linear MDPs (see Assumption A in [11] for a definition). The learner has access to multiple feature maps one of which is aligned with the true dynamics of the MDP.

Figure 2: CORRAL with $\epsilon$-Greedy bases with different exploration rates. [2]

**Theorem 4.4.** *Let $\mathcal{M} = (\mathcal{S}, \mathcal{A}, H, \mathbb{P}, r)$ be a linear MDP parametrized by an unknown feature map $\{\Phi^* : \mathcal{S} \times \mathcal{R} \to \mathbb{R}^d\}$. Let $\{\Phi_i(s,a)\}_{i=1}^M$ be a family of feature maps with $\Phi_i(s,a) \in \mathbb{R}^d$ and satisfying $\Phi^* \in \{\Phi_i(s,a)\}_{i=1}^M$. The regret of CORRAL with $\eta = \frac{M^{1/2}}{T^{1/2}d^{3/2}H^{3/2}}$ using LSVI-UCB base algorithms is: $R(T) \leq \tilde{\mathcal{O}}\left(\sqrt{Md^3H^3T}\right)$.*

We also observe that in practice, smoothing RL algorithms such as UCRL and PSRL and using a CORRAL master on top of them can lead to improved performance. A longer discussion is in Appendix A.

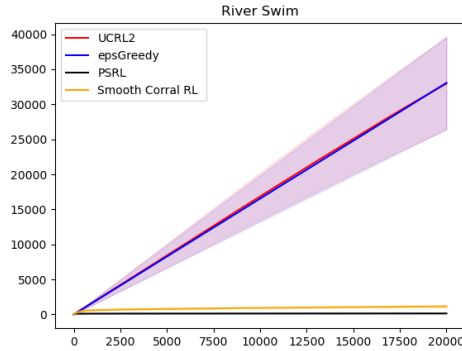

## 5 Base Smoothing

### 5.1 Non-increasing instantaneous regret

We introduce a two step "smoothing" procedure (Algorithm 3) which, given an algorithm $\mathcal{B}_j$ with concave (in $t$) cumulative high probability (see Definition 5.2) regret bound $U_j(t, \delta)$, constructs a smoothed algorithm $\tilde{\mathcal{B}}_j$ with an instantaneous regret bound $u_j(t, \delta) = U(t, \delta)/t$. If $U(t, \delta)$ is concave, $u(t, \delta)$ will be non-increasing in $t$.

Figure 3: $\epsilon$-Greedy vs UCRL2 vs PSRL in the River Swim environment [19].

Algorithm $\tilde{\mathcal{B}}_j$ works as follows. We have two steps in each round $s$. In step 1, we play $\mathcal{B}_j$. In step 2, at state $s$, we pick a time step $q$ in $[1, 2, .., s]$ uniformly at random, and re-play the policy made by $\mathcal{B}_j$ at time $q$. Since the policy of $\mathcal{B}_j$ at each round $[1, 2, ...s]$ is chosen with probability $1/s$ to be played at step 2, and $\mathcal{B}_j$ satisfies a high probability upper bound (Definition 5.1), the expected instantaneous regret of step 2 at round $s$ is at most $U(s, \delta)/s$ with high probability (Lemma D.1) which allows us to control term II in Eq. 2 via Theorem 5.2 (Appendix F.3). We use the superscript $(1)$ and $(2)$ to distinguish Step 1 and Step 2's action sets, policies, actions and rewards. Since the instantaneous regret of Step 2 is $1/s$ times the cumulative regret of Step 1, the cumulative regret of Step 2 over $S$ states is bounded roughly by $\sum_{s=1}^{S} 1/s \approx \log(S)$ times that of step 1.

**Algorithm 3** Smoothed Algorithm
---
**Input:** Base Algorithms $\mathcal{B}_j$; **Output:** Produce a smoothed base $\tilde{\mathcal{B}}_j$

Let $\pi_{s,j}$ be the policy of $\mathcal{B}_j$ in state $s$; Let $\tilde{\pi}_{s,j}^{(1)}, \tilde{\pi}_{s,j}^{(2)}$ be the policies of $\tilde{\mathcal{B}}_j$ in state $s$.
Initialize state counter $s = 1$.

  1: **for** $t = 1, \cdots, T$ **do**
  2:    **if** selected by master **then**

  3:

**Step 1**
> Receive action set $\mathcal{A}_t^{(1)} \sim D_S$
> Let $\tilde{\pi}_{s,i}^{(1)} = \pi_{s,i}$ from $\mathcal{B}_i$.
> Play action $a_{t,j}^{(1)} \sim \tilde{\pi}_{s,j}^{(1)}(\mathcal{A}_t^{(1)})$; Receive feedback $r_{t,j}^{(1)} = f(\mathcal{A}_t^{(1)}, \delta_{a_{t,j}^{(1)}}) + \xi_t^{(1)}$
> Calculate $\pi_{s+1,j}$ of $\mathcal{B}_j$ using $r_t^{(1)}$.

**Step 2**
> Receive action set $\mathcal{A}_t^{(2)} \sim D_S$.
> Sample $q \sim \text{Uniform}(0, \cdots, s)$; Let $\tilde{\pi}_{s,i}^{(2)} = \pi_{q,i}$ from $\mathcal{B}_i$.
> Play action $a_{t,j}^{(2)} \sim \tilde{\pi}_{s,j}^{(2)}(\mathcal{A}_t^{(2)})$; Receive feedback $r_{t,j}^{(2)} = f(\mathcal{A}_t^{(2)}, \delta_{a_{t,j}^{(2)}}) + \xi_t^{(2)}$

  4:      Send smoothed reward $r_{t,j}^{(2)}$ as both the rewards of Step 1 and Step 2 to the master.
  5:      $s \leftarrow s + 1$
  6:    **else**
  7:      **for** 2 steps **do**
  8:        Receive action set $\mathcal{A}_t \sim D_S$.
  9:        Choose action $a_{t,j}^{(2)} \sim \tilde{\pi}_{s,i}^{(2)}(\mathcal{A}_t)$.
10:      **end for**
11:    **end if**
12: **end for**
---

**Definition 5.1** $((U, \delta, T)-$Boundedness)**.** *Let* $U : \mathbb{R} \times [0,1] \to \mathbb{R}^+$. *We say an algorithm* $\mathcal{B}$ *is* $(U, \delta, T)-$*bounded if with probability at least* $1 - \delta$ *and for all rounds* $t \in [1, T]$, *the cumulative pseudo-regret is bounded above by* $U(t, \delta)$: $\sum_{l=1}^{t} f(\mathcal{A}_l, \pi^*) - f(\mathcal{A}_l, \pi_l) \leq U(t, \delta)$.

**Definition 5.2** $((U, \delta, \mathcal{T}^{(2)})-$Smoothness)**.** *Let* $U : \mathbb{R} \times [0,1] \to \mathbb{R}^+$. *We say a smoothed algorithm* $\tilde{\mathcal{B}}$ *is* $(U, \delta, \mathcal{T}^{(2)})-$*smooth if with probability* $1 - \delta$ *and for all rounds* $t \in [T]$, *the conditional expected instantaneous regret of Step 2 is bounded above by* $U(t, \delta)/t$:

$$\mathbf{E}_{\mathcal{A}_t \sim \mathcal{D}_S}[r_t^{(2)} | \mathcal{F}_{t-1}] \leq \frac{U(t, \delta)}{t}, \ \forall t \in [T]. \tag{3}$$

*Here* $\mathcal{F}_{t-1}$ *denotes the sigma algebra of all randomness up to the beginning of round* $t$.

In Appendix D we show that several algorithms such as UCB, LinUCB, $\epsilon$-greedy and EXP3 are $(U, \delta, T)$-bounded for appropriate functions $U$. In Propositon 5.1 we show that if $\mathcal{B}_j$ is bounded, then $\tilde{\mathcal{B}}_j$ is both bounded and smooth:

**Proposition 5.1.** *If* $U(t, \delta) > 8\sqrt{t \log(\frac{t^2}{\delta})}$, $\delta \leq \frac{1}{\sqrt{T}}$ *and* $\mathcal{B}_j$ *is* $(U, \delta, T)-$*bounded, then* $\tilde{\mathcal{B}}_j$ *is* $(6U \log(T), \delta, T)-$*bounded and* $(5U, \delta, \mathcal{T}^{(2)})-$*smooth.*

## 5.2 Regret Analysis

**Term I.** Note that we only send the smoothed reward of Step 2 to the master while the base plays and incurs regrets from both Step 1 and Step 2. We show in Appendix E that this does not affect the regret of the master significantly. For CORRAL with learning rate $\eta$, $\mathbf{E}[I] \leq O\left(\sqrt{MT} + \frac{M \ln T}{\eta} + T\eta\right) -$ $\frac{\mathbf{E}\left[\frac{1}{p_i}\right]}{40 \eta \ln T}$. For EXP3.P with exploration rate $p$, $\mathbf{E}[I] < O(\sqrt{MT} + \frac{1}{p} + MTp)$.

**Term II.** Term II is the regret of the base $i$ when it only updates its state when selected. We assume smoothed base algorithm $\tilde{\mathcal{B}}_i$ satisfies the smoothness and boundedness in Definitions 5.1 and 5.2. Note that when a smoothed base repeats its policy while not played, it repeats the next Step 2 policy (Algorithm 3) whose instantaneous regret is non-increasing.

Since the Step 2's conditional instantaneous regret (Definition 5.2) has a non-increasing upper bound, selecting $\tilde{\mathcal{B}}_i$ with probability $\underline{p}_i$ at every time step will result in the largest upper bound on its regret because the base is updated the least often. In this case the base will be updated every $1/\underline{p}_i$ time-steps and the regret upper bound will be roughly $\frac{1}{\underline{p}_i} U_i(T\underline{p}_i, \delta)$.

**Theorem 5.2.** *We have that* $\mathbf{E}[\text{II}] \leq \mathcal{O}\left(\mathbf{E}\left[\frac{1}{\underline{p}_i} U_i(T\underline{p}_i, \delta) \log T\right] + \delta T(\log T + 1)\right)$. *Here, the expectation is over the random variable* $\underline{p}_i$. *If* $U(t, \delta) = t^\alpha c(\delta)$ *for some* $\alpha \in [1/2, 1)$ *then,* $\mathbf{E}[\text{II}] \leq \tilde{\mathcal{O}}\left(T^\alpha c(\delta) \mathbf{E}\left[\frac{1}{\underline{p}_i^{1-\alpha}}\right] + \delta T(\log T + 1)\right)$.

**Total Regret.** Adding Term I and Term II gives us the following worst-case bound for CORRAL (maximized over $\underline{p}_i$ and with a chosen $\eta$) and EXP3.P (with a chosen $p$):

**Theorem 5.3.** *If a base algorithm is* $(U, \delta, T)$*-bounded for* $U(T, \delta) = T^\alpha c(\delta)$ *and some* $\alpha \in [1/2, 1)$ *and the choice of* $\delta = 1/T$, *the regret is upper bounded by:*

| | EXP3.P | CORRAL |
|---|---|---|
| *General* | $\tilde{O}\left(\sqrt{MT} + MTp + T^\alpha p^{\alpha-1} c(\delta)\right)$ | $\tilde{O}\left(\sqrt{MT} + \frac{M}{\eta} + T\eta + T\, c(\delta)^{\frac{1}{\alpha}} \eta^{\frac{1-\alpha}{\alpha}}\right)$ |
| *Known $\alpha$ Known $c(\delta)$* | $\tilde{O}\left(\sqrt{MT} + M^{\frac{1-\alpha}{2-\alpha}} T^{\frac{1}{2-\alpha}} c(\delta)^{\frac{1}{2-\alpha}}\right)$ | $\tilde{O}\left(\sqrt{MT} + M^\alpha T^{1-\alpha} + M^{1-\alpha} T^\alpha c(\delta)\right)$ |
| *Known $\alpha$ Unknown $c(\delta)$* | $\tilde{O}\left(\sqrt{MT} + M^{\frac{1-\alpha}{2-\alpha}} T^{\frac{1}{2-\alpha}} c(\delta)\right)$ | $\tilde{O}\left(\sqrt{MT} + M^\alpha T^{1-\alpha} + M^{1-\alpha} T^\alpha c(\delta)^{\frac{1}{\alpha}}\right)$ |

# 6 Lower bound

In stochastic environments, algorithms such as UCB can achieve logarithmic regret bounds. Our model selection procedure however has a $O(\sqrt{T})$ overall regret. In this section, we show that in general it is impossible to obtain a regret better than $\Omega(\sqrt{T})$ even when one base has 0 regret.

**Theorem 6.1.** *There exists an algorithm selection problem, such that the regret for any time $T$ is lower bounded by* $R(T) = \Omega\left(\frac{\sqrt{T}}{\log(T)}\right)$.

*Proof sketch.* The two base algorithms are constructed such that one base algorithm has 0 regret and the gap between the algorithms closes at a rate of $\Theta(1/(\sqrt{t}\log(t)))$. We show that at this rate, any master will have a constant probability of misidentifying the optimal algorithm even after observing infinite pulls. Hence the regret of the master is of order $\Omega\left(\sum_{t=1}^T 1/(\sqrt{t}\log(t))\right) = \tilde{\Omega}(\sqrt{T})$. $\square$

CORRAL needs knowledge of the best base's regret to achieve the same regret. The following lower bound shows that this requirement is unavoidable:

**Theorem 6.2.** *Let there be two base algorithms where the best base has regret* $\tilde{O}(T^x)$ *for some* $0 < x < 1$. *If we don't know $x$ and we don't know the reward of the best arm, then the regret of the master algorithm can be* $\Omega(T^y)$ *with* $y > x$.

*Proof sketch.* Let there be two base algorithms, and let $R_1$ and $R_2$ be their regrets incurred when called by the model selection strategy. If $R_1 = o(R_2)$, we can construct the bases such that they both have zero regret after the learner stops selecting them. Therefore their regret when running alone are $R_1$ and $R_2$, and the learner has regret of the same order as $R_2$, which is higher than the regret of the better base running alone ($R_1$). If however $R_1 \approx R_2$, since the learner doesn't know the optimal arm reward, we create another environment where the optimal arm reward is different, so that in the new environment the regrets are no longer equal. $\square$

## Acknowledgments and Disclosure of Funding

Csaba Szepesvári gratefully acknowledges funding from the Canada CIFAR AI Chairs Program, Amii and NSERC.

## Broader impact

The work does not present any foreseeable societal consequence.

## Footnotes

[2]The shaded areas around UCB and CORRAL are the std. The shaded areas around the $\epsilon$-greedy bases are 0.1 of std. For small $\epsilon$, $\epsilon$-greedy has a very high variance because it either commits to the optimal arm or the sub-optimal arm at the beginning, so plotting the whole $std$ would make the plot unreadable.

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
