[Supplementary Material]

# Supplement to "Model Selection in Contextual Stochastic Bandit Problems"

In Section A we briefly outline other direct applications of our results. In Section B we reproduce the CORRAL master and EXP3.P master algorithms. In Section D we present the proofs for Section 5.1 and show that algorithms such as UCB, $\epsilon-$greedy, LinUCB and EXP3 are $(U, T, \delta)-$bounded. In Sections E, F and G we present the proofs of the bound for Term I, Term II and the total regret, respectively. In Section H we show the proofs of the lower bounds in Section 6. In Section I we show the proofs of the regret bounds of the applications in Section 4.

## A  Other Applications

We outline briefly some other direct applications of our results. Similar to Section 4, we apply the smoothing procedure (Section 5) to all base algorithms before running them with the masters.

### A.1  Generalized Linear Bandits with Unknown Link Function

[14] study the generalized linear bandit model for the stochastic $k$-armed contextual bandit problem. In round $t$ and given context $x_t \in \mathbb{R}^{d \times k}$, the learner chooses arm $i_t$ and observes reward $r_t = \mu(x_{t,i_t}^\top \theta^*) + \xi_t$ where $\theta^* \in \mathbb{R}^d$ is an unknown parameter vector, $\xi_t$ is a conditionally zero-mean random variable and $\mu : \mathbb{R} \to \mathbb{R}$ is called the link function. [14] obtain the high probability regret bound $\tilde{O}(\sqrt{dT})$ where the link function is known. Suppose we have a set of link functions $\mathbb{L}$ that contains the true link function $\mu$. Since the target regret $\tilde{O}(\sqrt{dT})$ is known, we can run CORRAL with the algorithm in [14] with each link function in the set as a base algorithm. From Theorem 5.3, CORRAL will achieve regret $\tilde{O}(\sqrt{|\mathbb{L}|dT})$.

### A.2  Bandits with Heavy Tail

[18] study the linear stochastic bandit problem with heavy tail. If the reward distribution has finite moment of order $1 + \epsilon_*$, [18] obtain the high probability regret bound $\tilde{O}\left(T^{\frac{1}{1+\epsilon_*}}\right)$. We consider the problem when $\epsilon_* \in (0, 1]$ is unknown with a known lower bound $L$ where $L$ is a conservative estimate and $\epsilon_*$ could be much larger than $L$. To the best of our knowledge, we provide the first result when $\epsilon_*$ is unknown. We use the algorithms in [18] with value of $\epsilon_*$ in the grid $[b^{\log_b(L)}, ..., b^1, b^0]$ for some $0 < b < 1$ as base algorithms with $\eta = T^{-1/2}$ for CORRAL. A direct application of Theorem 5.3 yields regret $\tilde{O}\left(T^{1-0.5b\epsilon_*}\right)$. When $\epsilon_* = 1$ (as in the case of finite variance), $\tilde{O}\left(T^{1-0.5b\epsilon_*}\right)$ is close to $\tilde{O}\left(T^{0.5}\right)$ when $b$ is close to 1.

### A.3  Reinforcement Learning Experiment Details

In Figure 3, we present results for the model selection problem among distinct RL algorithms in the River Swim environment [19]. We use three different bases, $\epsilon-$greedy $Q-$learning with $\epsilon = .1$, Posterior Sampling Reinforcement Learning (PSRL), as described in [16] and UCRL2 as described in [10]. The implementation of these algorithms and the environment is taken from TabulaRL (https://github.com/iosband/TabulaRL), a popular benchmark suite for tabular reinforcement learning problems. Smooth Corral uses a CORRAL master algorithm with a learning rate $\eta = \frac{15}{\sqrt{T}}$, all base algorithms are smoothed using Algorithm 3. The curves for UCRL2, PSRL and $\epsilon-$greedy are all of their un-smoothed versions. Each experiment was repeated 10 times and we have reported the mean cumulative regret and shaded a region around them corresponding to $\pm.3$ the standard deviation across these 10 runs.

## B  Master Algorithms

### B.1  Original Corral

The original Corral algorithm [2] is reproduced below.

---

**Algorithm 4** Original Corral

---

**Input:** Base Algorithms $\{\mathcal{B}_j\}_{j=1}^M$, learning rate $\eta$.

Initialize: $\gamma = 1/T, \beta = e^{\frac{1}{\ln T}}, \eta_{1,j} = \eta, \rho_1^j = 2M, \underline{p}_1^j = \frac{1}{\rho_1^j}, p_1^j = 1/M$ for all $j \in [M]$.

Initialize all base algorithms.

    **for** $t = 1, \cdots, T$ **do**

        Receive context $x_t \sim \mathcal{D}$.

        Receive policy $\pi_{t,j}$ from $\mathcal{B}_j$ for all $j \in [M]$.

        Sample $i_t \sim p_t$.

        Play action $a_t \sim \pi_{t,i_t}(x_t)$.

        Receive feedback $r_t = f(x_t, \delta_{a_t}) + \xi_t$.

        Send feedback $\frac{r_t}{\overline{p}_{t,i_t}} \mathbf{1}\{j = i_t\}$ to $\mathcal{B}_j$ for all $j \in [M]$.

        Update $p_t, \eta_t$ and $\underline{p}_t$ to $p_{t+1}, \eta_{t+1}$ and $\underline{p}_{t+1}$ using $r_t$ via Corral-Update.

        **for** $j = 1, \cdots, M$ **do**

            Set $\rho_{t+1}^j = \frac{1}{\underline{p}_{t+1}^j}$

        **end for**

    **end for**

---

**Algorithm 5** Corral-Update

---

**Input:** learning rate vector $\eta_t$, distribution $p_t$, lower bound $\underline{p}_t$ and current loss $r_t$

**Output:** updated distribution $\pi_{t+1}$, learning rate $\eta_{t+1}$ and loss range $\rho_{t+1}$

Update $p_{t+1} = \text{Log-Barrier-OMD}(p_t, \frac{r_t}{p_{t,i_t}} \mathbf{e}_{i_t}, \eta_t)$.

Set $p_{t+1} = (1 - \gamma)p_{t+1} + \gamma \frac{1}{M}$.

    **for** $j = 1, \cdots, M$ **do**

        **if** $\underline{p}_t^j > p_{t+1}^j$ **then**

            Set $\underline{p}_{t+1}^j = \frac{p_{t+1}^j}{2}, \eta_{t+1,j} = \beta \eta_{t,i}$,

        **else**

            Set $\underline{p}_{t+1}^j = \underline{p}_t^j, \eta_{t+1,j} = \eta_{t,i}$.

        **end if**

    **end for**

Return $p_{t+1}, \eta_{t+1}$ and $\underline{p}_{t+1}$.

---

**Algorithm 6** Log-Barrier-OMD($p_t, \ell_t, \eta_t$)

---

**Input:** learning rate vector $\eta_t$, previous distribution $p_t$ and current loss $\ell_t$

**Output:** updated distribution $p_{t+1}$

Find $\lambda \in [\min_j \ell_{t,j}, \max_j \ell_{t,j}]$ such that $\sum_{j=1}^M \frac{1}{\frac{1}{p_t^i} + \eta_{t,j}(\ell_{t,j} - \lambda)} = 1$

Return $p_{t+1}$ such that $\frac{1}{p_{t+1}^j} = \frac{1}{p_t^j} + \eta_{t,j}(\ell_{t,j} - \lambda)$

---

## B.2  Corral Master

We reproduce our Corral master algorithm below.

---
**Algorithm 7** Corral Master
---
**Input:** Base Algorithms $\{\mathcal{B}_j\}_{j=1}^M$, learning rate $\eta$.

Initialize: $\gamma = 1/T, \beta = e^{\frac{1}{\ln T}}, \eta_{1,j} = \eta, \rho_1^j = 2M, \underline{p}_1^j = \frac{1}{\rho_1^j}, p_1^j = 1/M$ for all $j \in [M]$.

&emsp;**for** $t = 1, \cdots, T$ **do**
&emsp;&emsp;Sample $i_t \sim p_t$.
&emsp;&emsp;Receive feedback $r_t$ from base $\mathcal{B}_{i_t}$.
&emsp;&emsp;Update $p_t$, $\eta_t$ and $\underline{p}_t$ to $p_{t+1}$, $\eta_{t+1}$ and $\underline{p}_{t+1}$ using $r_t$ via Corral-Update.
&emsp;&emsp;**for** $j = 1, \cdots, M$ **do**
&emsp;&emsp;&emsp;Set $\rho_{t+1}^j = \frac{1}{\underline{p}_{t+1}^j}$
&emsp;&emsp;**end for**
&emsp;**end for**
---

## B.3  EXP3.P Master

We reproduce the EXP3.P algorithm (Figure 3.1 in [4]) below. In this formulation we use $\eta = 1, \gamma = 2\beta k$ and $p = \frac{\gamma}{k}$.

---
**Algorithm 8** EXP3.P Master
---
**Input:** Base Algorithms $\{\mathcal{B}_j\}_{j=1}^M$, exploration rate $p$.

Initialize: $p_1^j = 1/M$ for all $j \in [M]$.

&emsp;**for** $t = 1, \cdots, T$ **do**
&emsp;&emsp;Sample $i_t \sim p_t$.
&emsp;&emsp;Receive feedback $r_t$ from base $\mathcal{B}_{i_t}$.
&emsp;&emsp;Compute the estimated gain for each base $j$: $\tilde{r}_{t,j} = \frac{r_{t,j}\mathbf{1}_{i_t=j}+p/2}{p_{j,t}}$ and update the estimated cumulative gain $\tilde{R}_{j,t} = \sum_{s=1}^t \tilde{r}_{s,j}$.
&emsp;&emsp;**for** $j = 1, \cdots, M$ **do**
&emsp;&emsp;&emsp;$p_{t+1}^j = (1-p)\frac{\exp \tilde{R}_{j,t}}{\sum_{n=1}^M \exp \tilde{R}_{n,t}} + p$
&emsp;&emsp;**end for**
&emsp;**end for**
---

# C  Some useful lemmas

**Lemma C.1.** *If $U(t,\delta) = t^\beta c(\delta)$, for $0 \le \beta \le 1$ then:*

$$U(l,\delta) \le \sum_{t=1}^l \frac{U(t,\delta)}{t} \le \frac{1}{\beta}U(l,\delta)$$

*Proof.* The LHS follows immediately from observing $\frac{U(t,\delta)}{t}$ is decreasing as a function of $t$ and therefore $\sum_{t=1}^l \frac{U(t,\delta)}{t} \ge l\frac{U(l,\delta)}{l} = U(l,\delta)$. The RHS is a consequence of bounding the sum by the integral $\int_0^l \frac{U(t,\delta)}{t}dt$, substituting the definition $U(t,\delta) = t^\beta c(\delta)$ and solving it. □

**Lemma C.2.** *If $f(x)$ is a concave and doubly differentiable function on $x > 0$ and $f(0) \ge 0$ then $f(x)/x$ is decreasing on $x > 0$*

*Proof.* In order to show that $f(x)/x$ is decreasing when $x > 0$, we want to show that $\left(\frac{f(x)}{x}\right)' = \frac{xf'(x)-f(x)}{x^2} < 0$ when $x > 0$. Since $0f'(0) - f(0) \le 0$, we will show that $g(x) = xf'(x) - f(x)$ is a non-increasing function on $x > 0$. We have $g'(x) = xf''(x) \le 0$ when $x \ge 0$ because $f(x)$ is concave. Therefore $xf'(x) - f(x) \le 0f'(0) - f(0) \le 0$ for all $x \ge 0$, which completes the proof.

□

**Lemma C.3.** *For any $\Delta \le \frac{1}{4} : \mathrm{KL}(\frac{1}{2}, \frac{1}{2} - \Delta) \le 3\Delta^2$.*

*Proof.* By definition $kl(p, q) = p \log(p/q) + (1 - p) \log(\frac{1-p}{1-q})$, so

$$\mathrm{KL}\left(\frac{1}{2}, \frac{1}{2} - \Delta\right) = \frac{1}{2}\left(\log(\frac{1}{1 - 2\Delta}) + \log(\frac{1}{1 + 2\Delta})\right)$$

$$= \frac{1}{2}\log\left(\frac{1}{1 - 4\Delta^2}\right) = \frac{1}{2}\log\left(1 + \frac{4\Delta^2}{1 - 4\Delta^2}\right) \le \frac{2\Delta^2}{1 - 4\Delta^2} \le \frac{2\Delta^2}{\frac{3}{4}} \le 3\Delta^2$$

$\square$

# D  Additional discussion from Section 5.1

## D.1  Proof of Proposition 5.1

Note that in Step 2 we are replaying the decision of $\mathcal{B}_i$ at time $s$ learned from a sequence of contexts $\mathcal{A}_1^{(1)}, ..., \mathcal{A}_s^{(1)}$ to another context $\mathcal{A}_\ell^{(2)}$. Since the contexts are sampled i.i.d from the same distribution, in Lemma D.1 we will show that when we reuse the policy learned from a series of contexts $\mathcal{A}_1, ..., \mathcal{A}_t$ to another series of context $\mathcal{A}_1', ..., \mathcal{A}_t'$, the regret is multiplied only by a constant factor. We call the regret when using a policy learned from a series of context to another series of contexts "replay regret".

**Lemma D.1.** *Let $h$ be a generic history of algorithm $\mathcal{B}$ and $h(t)$ the history $h$ up to time $t$. If $\mathcal{A}_1, \cdots, \mathcal{A}_t$ are i.i.d. contexts from $\mathcal{D}$ and $\pi_1, \cdots, \pi_t$ is the sequence of policies used by $\mathcal{B}$ on these contexts, the "expected replay regret" $R(t, h)$ is:*

$$R(t, h) = \mathbb{E}_{\mathcal{A}_1', \cdots, \mathcal{A}_t'}\left[\sum_{l=1}^{t} f(\mathcal{A}_l', \pi^*) - f(\mathcal{A}_l', \pi_l)\right] \tag{4}$$

*Where $\mathcal{A}_1', \cdots, \mathcal{A}_t'$ are i.i.d. contexts from $\mathcal{D}$ independent conditional on $\mathcal{F}_t$, the sigma algebra capturing all that has occurred up to time $t$. If $\mathcal{B}$ is $(U, \delta, T)-$bounded, $\max_{x,\pi} |f(x, \pi)| \le 1$, $U(t, \delta) > 8\sqrt{t \log(\frac{t^2}{\delta})}$, and $\delta \le \frac{1}{\sqrt{T}}$, then $\mathcal{B}$'s expected replay regret satisfies: $R(t, h) \le 4U(t, \delta) + 2\delta t \le 5U(t, \delta)$.*

*Proof.* Consider the following two martingale difference sequences:

$$\{M_l^1 := f(\mathcal{A}_l, \pi^*) - f(\mathcal{A}_l'', \pi^*)\}_{l=1}^{t}$$
$$\{M_l^2 := f(\mathcal{A}_l'', \pi_l) - f(\mathcal{A}_l, \pi_l)\}_{l=1}^{t}$$

Since $\max\left(|M_l^1|, |M_l^2|\right) \le 2$ for all $t$, a simple use of Azuma-Hoeffding yields:

$$\mathbb{P}\left(\left|\sum_l M_l^i\right| \ge U(t, \delta)\right) \le \mathbb{P}\left(\left|\sum_l M_l^i\right| \ge \sqrt{8t \log\left(\frac{8t^2}{\delta}\right)}\right)$$

$$\le 2\exp\left(-\frac{8t \log(\frac{8t^2}{\delta})}{8t}\right)$$

$$= \frac{\delta}{4t^2}.$$

Summing over all $t$, and all $i \in \{1, 2\}$ and applying the union bound, using the fact that $\sum_{t=1}^{T} \frac{1}{t^2} < 2$ implies that for all $t$, with probability $1 - \delta$,

$$\left|\underbrace{\left(\sum_{l=1}^{t} f(\mathcal{A}_l, \pi^*) - \sum_{l=1}^{t} f(\mathcal{A}_l, \pi_l)\right)}_{\text{I}} - \underbrace{\left(\sum_{l=1}^{t} f(\mathcal{A}_l'', \pi^*) - \sum_{l=1}^{t} f(\mathcal{A}_l'', \pi_l)\right)}_{\text{II}}\right| \le 2U(t, 2\delta).$$

Since with probability $1 - \delta$ term I is upper bounded by $U(t, \delta)$ for all $t$ a simple union bound implies that with probability $1 - 2\delta$ term II is upper bounded by $U(t, \delta) + 2\sqrt{8t \log\left(\frac{8t^2}{\delta}\right)} \le 4U(t, \delta)$ for all $t$. The replay expected regret $R(t, h)$ can be upper bounded by: $(1 - 2\delta)4U(t, \delta) + 2\delta t \le 4U(t, \delta) + 2\delta t$. The result follows.

$\square$

*Proof of Proposition 5.1.* Since the conditional instantaneous regret on Step 2 of round $t$ equals the average replay regret of the type 1 steps up to $t$, Lemma D.1 implies $\mathbf{E}[r_t^{(2)}|\mathcal{F}_{t-1}] \le \frac{5U(t,\delta)}{t}$. The regret of step 1 is bounded by $U(t, \delta)$. The regret of step 2 is bounded by $\sum_{t=1}^{T} \frac{5U(t,\delta)}{t} = 5U(T, \delta) \log(T)$. Therefore total regret is bounded by $6U(T, \delta) \log(T)$

$\square$

## D.2  Applications of Proposition 5.1

We now show that several algorithms are $(U, \delta, T)$−bounded:

**Lemma D.2.** *Assuming that the noise $\xi_t$ is conditionally 1-sub-Gaussian, UCB is $(U, \delta, [T])$-bounded with $U(t, \delta) = O(\sqrt{tk} \log \frac{tk}{\delta})$.*

**Lemma D.3** (Theorem 3 in [1]). *LinUCB is $(U, \delta, T)$-bounded with $U(t, \delta) = O(d\sqrt{t} \log(1/\delta))$.*

**Lemma D.4** (Theorem 1 in [7]). *When $k$ is finite, LinUCB is $(U, \delta, T)$-bounded with $U(t, \delta) = O(\sqrt{dt} \log^3(kT \log(T)/\delta))$.*

**Lemma D.5.** *If $c = \frac{10K \log(\frac{1}{\delta})}{\Delta_*^2}$ where $\Delta_j$ is the gap between the optimal arm and arm $j$ and $\Delta_* = \min_j \Delta_j$, then $\epsilon$−greedy with $\epsilon_t = \frac{c}{t}$ satisfies a $(U, \delta, T)$−bounded for $\delta \le \frac{\Delta_*^2}{T^3}$: $U(t, \delta) = 16\sqrt{\log(\frac{1}{\delta})t}$ when $k = 2$ and $U(t, \delta) = 20 \left(k \log(\frac{1}{\delta}) \left(\sum_{j=2}^{k} \Delta_j\right)\right)^{1/3} t^{2/3}$ when $k > 2$*

**Lemma D.6** (Theorem 1 in [17]). *Exp3 is $(U, \delta, T)$−bounded where $U(t, \delta) = O(\sqrt{tk} \log \frac{tk}{\delta})$.*

### D.2.1  Proof of Lemma D.2

*Proof.* The regret of UCB is bounded as $\sum_{i:\Delta_i>0} \left(3\Delta_i + \frac{16}{\Delta_i} \log \frac{2k}{\Delta_i \delta}\right)$ (Theorem 7 of Abbasi-Yadkori et al. [1]) where $\Delta_i$ is the gap between arm $i$ and the best arm. By substituting the worst-case $\Delta_i$ in the regret bound, $U(T, \delta) = O(\sqrt{Tk} \log \frac{Tk}{\delta})$. $\square$

### D.2.2  Proof of Lemma D.5

In this section we show that epsilon greedy satisfies a high probability regret bound. We adapt the notation to this setup. Let $\mu_1, \cdots, \mu_K$ be the unknown means of the $K$ arms. Recall that at time $t$ the epsilon Greedy algorithm selects with probability $\epsilon_t = \min(c/t, 1)$ an arm uniformly at random, and with probability $1 - \epsilon_t$ it selects the arm whose empirical estimate of the mean is largest so far. Let $\hat{\mu}_j^{(t)}$ denote the empirical estimate of the mean of arm $j$ after using $t$ samples.

Without loss of generality let $\mu_1$ be the optimal arm. We denote the gaps as $\Delta_j = \mu_1 - \mu_j$ for all $j$. Let $\Delta_*$ be the smallest nonzero gap. We follow the discussion in [3] and start by showing that under the right assumptions, and for a horizon of size $T$, the algorithm satisfies a high probability regret bound for all $t \le T$. The objective of this section is to prove the following Lemma:

**Lemma D.7.** *If $c = \frac{10k \log(\frac{1}{\delta})}{\Delta_*^2}$[3], then $\epsilon$−greedy with $\epsilon_t = \frac{c}{t}$ is $(\delta, U, T)$−stable for $\delta \le \frac{\Delta_*^2}{T^3}$ and $U(t, \delta) = \frac{30k \log(\frac{1}{\delta})}{\Delta_*^2} \left(\sum_{j=2}^{k} \frac{\Delta_j}{\Delta_*^2} + \Delta_j\right) \log(t+1)$.*

*Proof.* Let $E(t) = \frac{1}{2K}\sum_{l=1}^{t}\epsilon_l$ and denote by $T_j(t)$ the random variable denoting the number of times arm $j$ was selected up to time $t$. We start by analyzing the probability that a suboptimal arm $j > 1$ is selected at time $t$:

$$\mathbb{P}(j \text{ is selected at time } t) \leq \frac{\epsilon_t}{k} + \left(1 - \frac{\epsilon_t}{k}\right)\mathbb{P}\left(\hat{\mu}_j^{(T_j(t))} \geq \hat{\mu}_1^{(T_1(t))}\right) \tag{5}$$

Let's bound the second term.

$$\mathbb{P}\left(\hat{\mu}_j^{(T_j(t))} \geq \hat{\mu}_1^{(T_1(t))}\right) \leq \mathbb{P}\left(\hat{\mu}_j^{(T_j(t))} \geq \mu_j + \frac{\Delta_j}{2}\right) + \mathbb{P}\left(\hat{\mu}_1^{(T_1(t))} \leq \mu_1 - \frac{\Delta_j}{2}\right)$$

The analysis of these two terms is the same. Denote by $T_j^R(t)$ the number of times arm $j$ was played as a result of a random epsilon greedy move. We have:

$$\mathbb{P}\left(\hat{\mu}_j^{(T_j(t))} \geq \mu_j + \frac{\Delta_j}{2}\right) = \sum_{l=1}^{t}\mathbb{P}\left(T_j(t) = l \text{ and } \hat{\mu}_j^{(l)} \geq \mu_j + \frac{\Delta_j}{2}\right)$$

$$= \sum_{l=1}^{t}\mathbb{P}\left(T_j(t) = l | \hat{\mu}_j^{(l)} \geq \mu_j + \frac{\Delta_j}{2}\right)\mathbb{P}\left(\hat{\mu}_j^{(l)} \geq \mu_j + \frac{\Delta_j}{2}\right)$$

$$\overset{I}{\leq} \sum_{l=1}^{t}\mathbb{P}\left(T_j(t) = l | \hat{\mu}_j^{(l)} \geq \mu_j + \frac{\Delta_j}{2}\right)\exp(-\Delta_j^2 t/2)$$

$$\overset{II}{\leq} \sum_{l=1}^{\lfloor E(t)\rfloor}\mathbb{P}\left(T_j(t) = l | \hat{\mu}_j^{(l)} \geq \mu_j + \frac{\Delta_j}{2}\right) + \frac{2}{\Delta_j^2}\exp(-\Delta_j^2\lfloor E(t)\rfloor/2)$$

$$\leq \sum_{l=1}^{\lfloor E(t)\rfloor}\mathbb{P}\left(T_j^R(t) = l | \hat{\mu}_j^{(l)} \geq \mu_j + \frac{\Delta_j}{2}\right) + \frac{2}{\Delta_j^2}\exp(-\Delta_j^2\lfloor E(t)\rfloor/2)$$

$$\leq \underbrace{\lfloor E(t)\rfloor\mathbb{P}\left(T_j(t)^R \leq \lfloor E(t)\rfloor\right)}_{(1)} + \underbrace{\frac{2}{\Delta_j^2}\exp(-\Delta_j^2\lfloor E(t)\rfloor/2)}_{(2)}$$

Inequality I is a consequence of a Chernoff bound. Inequality II follows because $\sum_{l=E+1}^{\infty}\exp(-\alpha l) \leq \frac{1}{a}\exp(-\alpha E)$. Term (1) corresponds to the probability that within the interval $[1, \cdots, t]$, the number of greedy pulls to arm $j$ is at most half its expectation. Term (2) is already "small".

Recall $\epsilon_t = \min(c/t, 1)$. Let $c = \frac{10K\log(T^3/\gamma)}{\Delta_*^2}$ for some $\gamma \in (0, 1)$ satisfying $\gamma \leq \Delta_j^2$. Under these assumptions we can lower bound $E(t)$: Indeed if $t \geq \frac{10K\log(T^3/\gamma)}{\Delta_*^2}$:

$$\frac{1}{2K}\sum_{l=1}^{t}\epsilon_l = \frac{5\log(T^3/\gamma)}{\Delta_*^2} + \frac{5\log(T^3/\delta)}{\Delta_*^2}\sum_{l=\log(T^3/\gamma)}^{t}\frac{1}{l}$$

$$\geq \frac{5\log(T^3/\gamma)}{\Delta_*^2} + \frac{5\log(T^3/\gamma)\log(t)}{2\Delta_*^2}$$

$$\geq \frac{5\log(T^3/\gamma)}{\Delta_*^2}$$

By Bernstein's inequality (see derivation of equation (13) in [3]) it is possible to show that :

$$\mathbb{P}\left(T_j^R(t) \leq E(t)\right) \leq \exp\left(-E(t)/5\right) \tag{6}$$

Hence for $t \geq \frac{10k\log(T^3/\gamma)}{\Delta_*^2}$:

$$\mathbb{P}\left(T_j^R(t) \leq E(t)\right) \leq \left(\frac{\gamma}{T^3}\right)^{\frac{1}{\Delta_*^2}}$$

And therefore since $E(t) \leq T$ and $\frac{1}{\Delta_*} \geq 1$ we can upper bound (1) as:

$$\lfloor E(t) \rfloor \mathbb{P} \left( T_j(t)^R \leq \lfloor E(t) \rfloor \right) \leq \left( \frac{\gamma}{T^2} \right)^{\frac{1}{\Delta_*^2}} \leq \frac{\gamma}{T^2}$$

Now we proceed with term (2):

$$\frac{2}{\Delta_j^2} \exp \left( -\Delta_j^2 \lfloor E(t) \rfloor / 2 \right) \leq \frac{2}{\Delta_j^2} \exp \left( -5k \log(\frac{T^3}{\gamma}) \frac{\Delta_j^2}{\Delta_*^2} \right)$$

$$\leq \frac{2}{\Delta_j^2} \exp \left( -5k \log(\frac{T^3}{\gamma}) \right)$$

$$= \frac{2}{\Delta_j^2} \left( \frac{\gamma}{T^3} \right)^{5k}$$

By the assumption $\gamma \leq \Delta_j^2$ the last term is upper bounded by $\frac{\gamma}{T^3}$.

The previous discussion implies that for $c = \frac{10k \log(T^3/\gamma)}{\Delta_*^2}$, the probability of choosing a suboptimal arm $j \geq 2$ at time $t$ for $t \geq \frac{10k \log(T^3/\gamma)}{\Delta_*^2}$ as a **greedy choice** is upper bounded by $2\frac{\gamma}{T}$. In other words after $t \geq \frac{10k \log(T^3/\gamma)}{\Delta_*^2}$, suboptimal arms with probability $1 - \frac{1}{T}$ over all $t$ are only chosen as a result of a exploration uniformly random epsilon greedy action.

A similar argument as the one that gave us Equation 6 can be used to upper bound the probability that at a round $t$, $T_j(t)^R$ be much larger than its mean:

$$\mathbf{P} \left( T_j^R(t) \geq 3E(j) \right) \leq \exp(-E(t)/5)$$

We can conclude that with probability more than $1 - \frac{k\gamma}{T}$ and for all $t$ and arms $j$, $T_j^R(t) \leq 3E(t)$. Combining this with the obsevation that after $t \geq \frac{10k \log(T^3/\gamma)}{\Delta_*^2}$ and with probability $1 - \frac{k\gamma}{T}$ over all $t$ simultaneously (by union bound) regret is only incurred by random exploration pulls (and not greedy actions), we can conclude that with probability $1 - \frac{2k\gamma}{T}$ simultaneously for all $t \geq \frac{10k \log(T^3/\gamma)}{\Delta_*^2}$ the regret incurred is upper bounded by:

$$\underbrace{\frac{10k \log(T^3/\gamma)}{\Delta_*^2} \cdot \frac{1}{k} \sum_{j=2}^{k} \Delta_j}_{I} + \underbrace{3E(t) \sum_{j=2}^{k} \Delta_j}_{II}$$

Where $I$ is a crude upper bound on the regret incurred in the first $\frac{10k \log(T^3/\gamma)}{\Delta_*^2}$ rounds and $II$ is an upper bound for the regret incurred in the subsequent rounds.

Since $E(t) \leq \frac{20k \log(T^3/\gamma)}{\Delta_*^2} \log(t)$ we can conclude that with probability $1 - \frac{2k\gamma}{T}$ for all $t \leq T$ the cumulative regret of epsilon greedy is upper bounded by $f(t) = 30K \log(T^3/\gamma) \left( \sum_{j=2}^{k} \frac{\Delta_j}{\Delta_*^2} + \Delta_j \right) \max(\log(t), 1)$, the result follows by identifying $\delta = \gamma/T^3$.

$\square$

We now show the proof of Lemma D.5 the instance-independent regret bound for $\epsilon$-greedy:

**Lemma D.8** (Lemma D.5). *If $c = \frac{10k \log(\frac{1}{\delta})}{\Delta_*^2}$, then $\epsilon$−greedy with $\epsilon_t = \frac{c}{t}$ is $(\delta, U, T)$−stable for $\delta \leq \frac{\Delta_*^2}{T^3}$ and:*

1. *$U(t, \delta) = 16 \sqrt{\log(\frac{1}{\delta})t}$ when $k = 2$.*

2. *$U(t, \delta) = 20 \left( K \log(\frac{1}{\delta}) \left( \sum_{j=2}^{K} \Delta_j \right) \right)^{1/3} t^{2/3}$ when $k > 2$.*

*Proof.* Let $\Delta$ be some arbitrary gap value. Let $R(t)$ denote the expected regret up to round $t$. We recycle the notation from the proof of Lemma D.7, recall $\delta = \gamma/T^3$.

$$R(t) = \sum_{\Delta_j \leq \Delta} \Delta_j \mathbb{E}\left[T_j(t)\right] + \sum_{\Delta_j \geq \Delta} \Delta_j \mathbb{E}\left[T_j(t)\right]$$

$$\leq \Delta t + \sum_{\Delta_j \geq \Delta} \Delta_j \mathbb{E}\left[T_j(t)\right]$$

$$\leq \Delta t + 30k \log(T^3/\gamma) \left( \sum_{\Delta_j \geq \Delta}^{k} \frac{\Delta_j}{\Delta_*^2} + \Delta_j \right) \log(t)$$

$$\leq \Delta t + 30k \log(T^3/\gamma) \left( \sum_{\Delta_j \geq \Delta}^{k} \frac{\Delta_j}{\Delta_*^2} \right) + 30k \log(T^3/\gamma) \log(t) \left( \sum_{\Delta_j \geq \Delta}^{k} \Delta_j \right) \quad (7)$$

When $K = 2$, $\Delta_2 = \Delta_*$ and therefore (assuming $\Delta < \Delta_2$):

$$R(t) \leq \Delta t + \frac{30k \log(T^3/\gamma)}{\Delta_2} + 30k \log(T^3/\gamma) \log(t) \Delta_2$$

$$\leq \Delta t + \frac{30k \log(T^3/\gamma)}{\Delta} + 30k \log(T^3/\gamma) \log(t) \Delta_2$$

$$\overset{\text{I}}{\leq} \sqrt{30k \log(T^3/\gamma)t} + 30k \log(T^3/\gamma) \log(t) \Delta_2$$

$$\overset{\text{II}}{\leq} 8\sqrt{k \log(T^3/\gamma)t}$$

$$\leq 16\sqrt{\log(T^3/\gamma)t}$$

Inequality I follows from setting $\Delta$ to the optimizer, which equals $\Delta = \sqrt{\frac{30k \log(T^3/\gamma)}{t}}$. The second inequality II is satisfied for $T$ large enough. We choose this expression for simplicity of exposition.

When $K > 2$ notice that we can arrive to a bound similar to 7:

$$R(t) \leq \Delta t + 30k \log(T^3/\gamma) \left( \sum_{\Delta_j \geq \Delta}^{k} \frac{\Delta_j}{\Delta^2} \right) + 30k \log(T^3/\gamma) \log(t) \left( \sum_{\Delta_j \geq \Delta}^{k} \Delta_j \right)$$

Where $\Delta_*$ is substituted by $\Delta$. This can be obtained from Lemma D.7 by simply substituting $\Delta_*$ with $\Delta$ in the argument for arms $j : \Delta_j \geq \Delta$.

We upper bound $\sum_{\Delta_j \geq \Delta} \Delta_j$ by $\sum_{j=2}^{k} \Delta_j$. Setting $\Delta$ to the optimizer of the expression yields $\Delta = \left( \frac{30k \log(T^3/\gamma)\left(\sum_{j=2}^{k} \Delta_j\right)}{t} \right)^{1/3}$, and plugging this back into the equation we obtain:

$$R(t) \leq 2 \left( 30k \log(T^3/\gamma) \left( \sum_{j=2}^{k} \Delta_j \right) \right)^{1/3} t^{2/3} + 30k \log(T^3/\gamma) \log(t) \left( \sum_{j=2}^{k} \Delta_j \right)$$

$$\overset{\xi}{\leq} 20 \left( k \log(T^3/\gamma) \left( \sum_{j=2}^{k} \Delta_j \right) \right)^{1/3} t^{2/3}$$

The inequality $\xi$ is true for $T$ large enough. We choose this expression for simplicity of exposition.

$\square$

# E Bounding term I

When the base algorithms are not chosen, they repeat their step 2's policy to ensure that the conditional instantaneous regret is decreasing. Therefore when the base algorithms are chosen by the master,

we must also only send step 2's rewards to the master as feedback signals. This is to ensure that the rewards of the bases at time $t$ do not depend on whether they are selected by the master at time $t$. However, since the bases play and incur regrets from both step 1 and step 2 when they are chosen, we must account to the difference between the reward of step 1 and step 2 (that the bases incur when they play the arms), and 2 times the reward of step 2 (that the bases send to the master as feedback signals).

Since we assume all base algorithms to be smoothed and satisfy a two step feedback structure, we also denote by $\pi_t^{(j)}$ as the policy used by the master during round $t$, step $j$. Term I, the regret of the master with respect to base $i$ can be written as:

$$\mathbf{E}\left[\mathrm{I}\right] = \mathbf{E}\left[\sum_{t=1}^{T}\sum_{j=1}^{2} f(\mathcal{A}_t^{(j)}, \pi_{t,i}^{(j)}) - f(\mathcal{A}_t^{(j)}, \pi_t^{(j)})\right] \tag{8}$$

Recall that the master algorithm is updated only using the reward of Step 2 of base algorithms even though the bases play both step 1 and 2. Let $\mathbb{T}_i$ is the random subset of rounds when $\mathcal{M}$ choose base $\mathcal{B}_i$, ($i_t = i$). Adding and subtracting terms $\{f(\mathcal{A}_t^{(1)}, \pi_t^{(2)})\}_{t=1}^{T}$ we see that:

$$\mathrm{I} = \sum_{t=1}^{T}\sum_{j=1}^{2} f(\mathcal{A}_t^{(j)}, \pi_{t,i}^{(j)}) - f(\mathcal{A}_t^{(j)}, \pi_t^{(j)})$$

$$= \underbrace{\sum_{t \in \mathbb{T}_i}\sum_{j=1}^{2} f(\mathcal{A}_t^{(j)}, \pi_{t,i}^{(j)}) - f(\mathcal{A}_t^{(j)}, \pi_t^{(j)})}_{\mathrm{I}_0} + \underbrace{\sum_{t \in \mathbb{T}_i^c}\sum_{j=1}^{2} f(\mathcal{A}_t^{(j)}, \pi_{t,i}^{(j)}) - f(\mathcal{A}_t^{(j)}, \pi_t^{(j)})}_{\mathrm{I}_1}$$

$$\overset{(i)}{=} \underbrace{\sum_{t \in \mathbb{T}_i}\sum_{j=1}^{2} f(\mathcal{A}_t^{(j)}, \pi_{t,i}^{(2)}) - f(\mathcal{A}_t^{(j)}, \pi_t^{(2)})}_{\mathrm{I}_0'} + \underbrace{\sum_{t \in \mathbb{T}_i^c}\sum_{j=1}^{2} f(\mathcal{A}_t^{(j)}, \pi_{t,i}^{(2)}) - f(\mathcal{A}_t^{(j)}, \pi_t^{(j)})}_{\mathrm{I}_1'}$$

$$\overset{(ii)}{=} \underbrace{\sum_{t=1}^{T}\sum_{j=1}^{2} f(\mathcal{A}_t^{(j)}, \pi_{t,i}^{(2)}) - f(\mathcal{A}_t^{(j)}, \pi_t^{(2)})}_{\mathrm{I}_A} + \underbrace{\sum_{t \in \mathbb{T}_i^c} f(\mathcal{A}_t^{(1)}, \pi_t^{(2)}) - f(\mathcal{A}_t^{(1)}, \pi_t^{(1)})}_{\mathrm{I}_B}$$

Equality $(i)$ holds because term $\mathrm{I}_0$ equals zero and therefore $\mathrm{I}_0 = \mathrm{I}_0'$ and in all steps $t \in \mathbb{T}_i^c$, base $i$ repeated a policy of step 2 so that $\mathrm{I}_1 = \mathrm{I}_1'$. Equality $(ii)$ follows from adding and subtracting term $\mathrm{I}_B$. Term $\mathbf{E}\left[\mathrm{I}_A\right]$ is the regret of the master with respect to base $i$. Term $\mathbf{E}\left[\mathrm{I}_B\right]$ accounts for the difference between the rewards of step 1 and step 2 (that the bases incur) and 2 times the rewards of step 2 (that the bases send to the master). We now focus on bounding $\mathbf{E}\left[\mathrm{I}_A\right]$ and $\mathbf{E}\left[\mathrm{I}_B\right]$.

**Modified step $2$'s rewards.** We introduce the following small modification to the algorithm's feedback. This will become useful to control $\mathbf{E}\left[I_B\right]$. Instead of sending the master the unadulterated $r_{t,j}^{(2)}$ feedback, at all time step $t$, all bases will send the following modified feedback:

$$r_{t,j}^{(2)'} = r_{t,j}^{(2)} - \frac{U(s_{t,j}, \delta)}{s_{t,j}} \tag{9}$$

This reward satisfies:

$$\mathbf{E}\left[r_{t,j}^{(2)'} | \mathcal{F}_{t-1}\right] = \mathbf{E}\left[f(\mathcal{A}_t^{(2)}, \pi_t^{(2)}) | \mathcal{F}_{t-1}\right] - \frac{U_j(s_{t,j}, \delta)}{s_{t,j}}$$

We'll show that this modification allows us to control term $I_B$ in Section E.2. Since this modification is performed internally by all bases, we note that term $I_A$ corresponds to an adversarial master that is always fed modified rewards from all bases and trying to compete against base $i$ also with modified

rewards. Therefore any worst case bound of term $I_A$ of an adversarial master will not be affected by this modification of the reward sequence of all bases.

Term $I_B$ is the difference between the (modified) rewards of step 2 and step 1 which, due to the introduced modification, should intuitively be small because the cumulative (modified) rewards of step 2 are designed be smaller than step 1. In section E.2 we show that $\mathbb{E}\left[I_B\right] \leq 8\sqrt{MT \log(\frac{4TM}{\delta})}$. Therefore $\mathbf{E}\left[I\right] \leq \mathbf{E}\left[I_A\right] + 8\sqrt{MT \log(\frac{4TM}{\delta})}$ .

Since any base $j$ sends the modified reward to the master when it is chosen, when it is not chosen and repeats its step 2's policy, the reward also needs to be modified in the same way as in Equation 9. This is to ensure that the rewards of the base at time $t$ do not depend on whether it is selected by the master at time $t$. We now discuss how this modification affects term II. Note that the modification increases term II (which only depends on base $i$) at each time step $t$ by $\frac{U_i(s_{t,i},\delta)}{s_{t,i}}$. Since the original instantaneous regret of base $i$ at step 2 is bounded by a term of the same order, the modification increases term II by only a constant factor (Section F).

## E.1   Bounding $\mathbf{E}\left[I_A\right]$

As we explain above, since the modification of the bases' rewards in Equation 9 is internal within the bases, and the master is a $k$-armed bandit adversarial algorithm, the worst-case performance of the master against any adversarial sequence of rewards will not be affect when the sequence of rewards of the bases changes.

### E.1.1   CORRAL Master

Notice that:

$$\mathbf{E}\left[I_A\right] = \mathbf{E}\left[\sum_{t=1}^{T} 2f(\mathcal{A}_t^{(2)}, \pi_{t,i}^{(2)}) - 2f(\mathcal{A}_t^{(2)}, \pi_t^{(2)})\right]$$

We can easily bound this term using Lemma 13 from [2]. Indeed, in term $I_A$, the policy choice for all base algorithms $\{\tilde{\mathcal{B}}_m\}_{m=1}^{M}$ during any round $t$ is chosen before the value of $i_t$ is revealed. This ensures the estimates $\frac{2r_t^{(2)}}{p_t^{i_t}}$ and 0 for all $i \neq i_t$ are indeed unbiased estimators of the base algorithm's rewards.

We conclude:

$$\mathbf{E}\left[I_A\right] \leq O\left(\frac{M \ln T}{\eta} + T\eta\right) - \frac{\mathbf{E}\left[\frac{1}{p_i}\right]}{40\eta \ln T}$$

### E.1.2   EXP3.P Master

Since $\mathbf{E}\left[I_A\right]$ is the regret of base $i$ with respect to the master, it can be upper bounded by the $k$-armed bandit regret of the master with $M$ arms. Choose $\eta = 1, \gamma = 2k\beta$ in Theorem 3.3 in [4], we have that if $p \leq \frac{1}{2k}$, the regret of EXP3.P:

$$\mathbf{E}\left[I_A\right] \leq \tilde{O}\left(MTp + \frac{\log(k\delta^{-1})}{p}\right)$$

## E.2 Bounding $\mathbf{E}\left[\mathrm{I}_B\right]$

Notice that:

$$\mathbf{E}\left[\mathrm{I}_B\right] = \mathbf{E}\left[\sum_{t\in\mathbb{T}_i^c} f(\mathcal{A}_t^{(1)}, \pi_t^{(2)}) - f(\mathcal{A}_t^{(1)}, \pi_t^{(1)})\right]$$

$$= \mathbf{E}\left[\underbrace{\sum_{t\in\mathbb{T}_i^c} f(\mathcal{A}_t^{(2)}, \pi_t^{(2)}) - f(\mathcal{A}_t^{(1)}, \pi_t^{(1)})}_{\mathrm{I}_B'}\right]$$

$$= \mathbf{E}\left[\sum_{t\in\mathbb{T}_i^c} f(\mathcal{A}_t^{(2)}, \pi_t^{(2)}) - f(\mathcal{A}_t^{(2)}, \pi^*) + f(\mathcal{A}_t^{(2)}, \pi^*) - f(\mathcal{A}_t^{(1)}, \pi_t^{(1)})\right]$$

$$= \mathbf{E}\left[\sum_{t\in\mathbb{T}_i^c} f(\mathcal{A}_t^{(2)}, \pi_t^{(2)}) - f(\mathcal{A}_t^{(2)}, \pi^*) + f(\mathcal{A}_t^{(1)}, \pi^*) - f(\mathcal{A}_t^{(1)}, \pi_t^{(1)})\right]$$

In order to bound this term we will make an extra assumption.

**Assumption A1** *(**Bounded Expected Rewards**)* We assume $|f(\mathcal{A}, \pi)| \leq 1$ for all $\mathcal{A}$ and all policies $\pi$.

Substituting the modified step 2 rewards in Equation 9 back into the expectation for $\mathbf{E}\left[\mathrm{I}_B\right]$ becomes:

$$\mathbb{E}[\mathrm{I}_B] = \mathbf{E}\left[\sum_{t\in\mathbb{T}_i^c} f(\mathcal{A}_t^{(2)}, \pi_t^{(2)}) - f(\mathcal{A}_t^{(2)}, \pi^*) - \frac{U_{j_t}(s_{t,j_t}(t), \delta)}{s_{t,j_t}} + f(\mathcal{A}_t^{(1)}, \pi^*) - f(\mathcal{A}_t^{(1)}, \pi_t^{(1)})\right]$$

$$= \sum_{j\neq i} \mathbf{E}\left[\sum_{t\in\mathbb{T}_j} f(\mathcal{A}_t^{(2)}, \pi_{t,j}^{(2)}) - f(\mathcal{A}_t^{(2)}, \pi^*) - \frac{U_j(s_{t,j}, \delta)}{s_{t,j}} + f(\mathcal{A}_t^{(1)}, \pi^*) - f(\mathcal{A}_t^{(1)}, \pi_{t,j}^{(1)})\right]$$

$$\overset{(1)}{\leq} \sum_{j\neq i} \mathbf{E}\left[\sum_{t\in\mathbb{T}_j} f(\mathcal{A}_t^{(2)}, \pi_{t,j}^{(2)}) - f(\mathcal{A}_t^{(2)}, \pi^*) + f(\mathcal{A}_t^{(1)}, \pi^*) - f(\mathcal{A}_t^{(1)}, \pi_{t,j}^{(1)})\right] - U_j(s_{T,j}, \delta) \tag{10}$$

Inequality (1) follows because by Lemma C.1 applied to $U_j(t, \delta)$.

Observe that if the $j-$th algorithm was in its $U_j$-compatible environment (also referred to as its "natural environment"), then for any instantiation of $\mathbb{T}_j$ and with high probability:

$$\left(\sum_{t\in\mathbb{T}_j} f(\mathcal{A}_t^{(2)}, \pi_{t,j}^{(2)}) - f(\mathcal{A}_t^{(2)}, \pi^*) + f(\mathcal{A}_t^{(1)}, \pi^*) - f(\mathcal{A}_t^{(1)}, \pi_{t,j}^{(1)})\right) - U_j(T_j(T), \delta) \leq$$

$$\left(\sum_{t\in\mathbb{T}_j} f(\mathcal{A}_t^{(1)}, \pi^*) - f(\mathcal{A}_t^{(1)}, \pi_{t,j}^{(1)})\right) - U_j(T_j(T), \delta) \leq 0 \tag{11}$$

The first inequality follows because by definition $f(\mathcal{A}_t^{(2)}, \pi^*) \geq f(\mathcal{A}_t^{(2)}, \pi_t^{(2)})$ and the last because of the high probability regret bound satisfied by $\mathcal{B}_j$.

When $\mathcal{B}_j$ is not in its $U_j$-compatible environment, this condition may or may not be violated. If this condition is violated, we need to make sure $\mathcal{B}_j$ is dropped by the master. Since it is impossible to compute the terms $f(\mathcal{A}_t^{(2)}, \pi_t^{(2)}) - f(\mathcal{A}_t^{(2)}, \pi^*)$ and $f(\mathcal{A}_t^{(1)}, \pi^*) - f(\mathcal{A}_t^{(1)}, \pi_t^{(1)})$ directly, we instead rely on the following test:

**Base Test.** Let $\mathbb{T}_j(l)$ be the first set of $l$ indices when the master chose to play base $j$. If at any point during the history of the algorithm we encounter

$$\sum_{t \in \mathbb{T}_j(l)} r_{t,j}^{(2)} - r_{t,j}^{(1)} > U_j(T_j(T), \delta) + 2\sqrt{2l \log\left(\frac{4TM}{\delta}\right)} \tag{12}$$

Then we drop base $\mathcal{B}_j$.

The logic of this step comes from a simple Azuma-Hoeffding martingale bound along with Assumption A1 with probability at least $1 - \delta/M$ and for all $l \in [T]$:

$$\left| \sum_{\ell=1}^{l} f(\mathcal{A}_\ell^{(2)}, \pi^*) - f(\mathcal{A}_\ell^{(1)}, \pi^*) \right| \leq \sqrt{2l \log\left(\frac{4TM}{\delta}\right)} \tag{13}$$

$$\left| \sum_{\ell=1}^{l} r_{\ell,j}^{(2)} - r_{\ell,j}^{(1)} - f(\mathcal{A}_\ell^{(2)}, \pi_{\ell,j}^{(2)}) - f(\mathcal{A}_\ell^{(1)}, \pi_{\ell,j}^{(1)}) \right| \leq \sqrt{2l \log\left(\frac{4TM}{\delta}\right)} \tag{14}$$

This means that whenever $\mathcal{B}_j$ is in its $U_j$-compatible environment, combining Equation 10, with Equation 13 and Equation 14 we get, with probability at least $1 - \delta$:

$$\left| \left( \sum_{t \in \mathbb{T}_j} r_{t,j}^{(2)} - r_{t,j}^{(1)} \right) - \left( \sum_{t \in \mathbb{T}_j} f(\mathcal{A}_t^{(2)}, \pi_{t,j}^{(2)}) - f(\mathcal{A}_t^{(2)}, \pi^*) + f(\mathcal{A}_t^{(1)}, \pi^*) - f(\mathcal{A}_t^{(1)}, \pi_{t,j}^{(1)}) \right) \right|$$
$$\leq 2\sqrt{2l \log\left(\frac{4TM}{\delta}\right)}$$

Plugging in inequality 11, we conclude that if $\mathcal{B}_j$ is in its $U_j$-compatible environment with probability at least $1 - \delta$ for all $l \in [T]$:

$$\sum_{t \in \mathbb{T}_j} r_{t,j}^{(2)} - r_{t,j}^{(1)} \leq U_j(s_{T,j}, \delta) + 2\sqrt{2l \log\left(\frac{4TM}{\delta}\right)}$$

Therefore the violation of condition in Equation 12, means $\mathcal{B}_j$ couldn't have possibly been in its $U_j$-compatible environment. Furthermore, notice that in case Equation 12 holds (even if $\mathcal{B}_j$ is not in its $U_j$-compatible environment), then with probability at least $1 - \delta/M$:

$$\sum_{t \in \mathbb{T}_j} f(\mathcal{A}_t^{(2)}, \pi_{t,j}^{(2)}) - f(\mathcal{A}_t^{(2)}, \pi^*) + f(\mathcal{A}_t^{(1)}, \pi^*) - f(\mathcal{A}_t^{(1)}, \pi_{t,j}^{(1)}) \leq U_j(s_{T,j}, \delta) + 4\sqrt{2|\mathbb{T}_j| \log(4TM)} \tag{15}$$

Consequently, this test guarantees condition 15 is satisfied with for all $j \in [M]$ and with probability at least $1 - \delta$, thus implying:

$$\mathbf{E}[I_B] \leq \sum_{j \neq i} 4\sqrt{2|\mathbb{T}_j| \log(4TM)} \leq 8\sqrt{MT \log\left(\frac{4TM}{\delta}\right)}$$

The last inequality holds because $\sum_{i \neq j} \sqrt{|\mathbb{T}_j|} \leq \sqrt{TM}$.

# F   Bounding term II

Recall term II equals:

$$\mathbf{E}[\text{II}] = \mathbf{E}\left[ \sum_{t=1}^{T} f(\mathcal{A}_t, \pi^*) - f(\mathcal{A}_t, \pi_{s_{t,i,i}}) \right] \tag{16}$$

We use $n_t^i$ to denote the number of rounds base $i$ is chosen up to time $t$. Let $t_{l,i}$ be the round index of the $l-$th time the master chooses algorithm $\mathcal{B}_i$ and let $b_{l,i} = t_{l,i} - t_{l-1,i}$ with $t_{0,i} = 0$ and $t_{n_T^i+1,i} = T+1$. Let $\mathbb{T}_i \subset [T]$ be the set of rounds where base $i$ is chosen and $\mathbb{T}_i^c = [T]\backslash\mathbb{T}_i$. For $S \subset [T]$ and $j \in \{1,2\}$, we define the regret of the $i-$th base algorithm during Step $j$ of rounds $S$ as $R_i^{(j)}(S) = \sum_{t \in S} f(\mathcal{A}_t^{(j)}, \pi^*) - f(\mathcal{A}_t^{(j)}, \pi_{t,i}^{(j)})$. The following decomposition of $\mathbf{E}[\text{II}]$ holds:

$$\mathbf{E}[\text{II}] = \mathbf{E}\left[ R_i^{(1)}(\mathbb{T}_i) + \underbrace{R_i^{(2)}(\mathbb{T}_i) + R_i^{(1)}(\mathbb{T}_i^c) + R_i^{(2)}(\mathbb{T}_i^c)}_{\text{II}_0} \right]. \tag{17}$$

$R_i^{(1)}(\mathbb{T}_i)$ consists of the regret when base$-i$ was updated in step 1 while the remaining 3 terms consists of the regret when the policies are reused by step 2.

### F.1 Modified step $2$'s rewards

Note that we modified the rewards of step 2 as defined in Equation 9, both when the base is chosen and not chosen. We now analyze the effect of the modification:

$R(T)$

$$= \mathbf{E}\left[ \sum_{t=1}^{T}\sum_{j=1}^{2} f(\mathcal{A}_t^{(j)}, \pi^*) - f(\mathcal{A}_t^{(j)}, \pi_t^{(j)}) \right]$$

$$= \mathbf{E}\left[ \underbrace{\sum_{t=1}^{T}\sum_{j=1}^{2} f(\mathcal{A}_t^{(j)}, \pi_{s_{t,i},i}^{(j)}) - f(\mathcal{A}_t^{(j)}, \pi_t^{(j)})}_{\text{I}} \right] + \mathbf{E}\left[ \underbrace{\sum_{t=1}^{T}\sum_{j=1}^{2} f(\mathcal{A}_t^{(j)}, \pi^*) - f(\mathcal{A}_t^{(j)}, \pi_{s_{t,i},i}^{(j)})}_{\text{II}} \right]$$

$$= \mathbf{E}\left[ \sum_{t=1}^{T}\sum_{j=1}^{2} \left( f(\mathcal{A}_t^{(j)}, \pi_{s_{t,i},i}^{(j)}) - \mathbf{1}(t \in \mathbb{T}_i^c \text{ or } j = 2)\frac{U_i(s_{t,i}, \delta)}{s_{t,i}} \right) - f(\mathcal{A}_t^{(j)}, \pi_t^{(j)}) \right]$$

$$+ \mathbf{E}\left[ \sum_{t=1}^{T}\sum_{j=1}^{2} f(\mathcal{A}_t^{(j)}, \pi^*) - \left( f(\mathcal{A}_t^{(j)}, \pi_{s_{t,i},i}^{(j)}) - \mathbf{1}(t \in \mathbb{T}_i^c \text{ or } j = 2)\frac{U_i(s_{t,i}, \delta)}{s_{t,i}} \right) \right]$$

$$\leq \mathbf{E}\left[ \underbrace{\sum_{t=1}^{T}\sum_{j=1}^{2} \left( f(\mathcal{A}_t^{(j)}, \pi_{s_{t,i},i}^{(j)}) - \mathbf{1}(t \in \mathbb{T}_i^c \text{ or } j = 2)\frac{U_i(s_{t,i}, \delta)}{s_{t,i}} \right) - \left( f(\mathcal{A}_t^{(j)}, \pi_t^{(j)}) - \frac{U_{j_t}(s_{t,j_t}, \delta)}{s_{t,j_t}} \right)}_{\text{I}-\text{modified}} \right]$$

$$+ \mathbf{E}\left[ \underbrace{\sum_{t=1}^{T}\sum_{j=1}^{2} f(\mathcal{A}_t^{(j)}, \pi^*) - \left( f(\mathcal{A}_t^{(j)}, \pi_{s_{t,i},i}^{(j)}) - \mathbf{1}(t \in \mathbb{T}_i^c \text{ or } j = 2)\frac{U_i(s_{t,i}, \delta)}{s_{t,i}} \right)}_{\text{II}-\text{modified}} \right]$$

We provided a bound for term I-modified in Section E. In this section we concern ourselves with II$-$modified. Notice its expectation can be written as:

$$\mathbf{E}[\text{II} - \text{modified}] = \mathbf{E}[\text{II}] + \mathbf{E}\left[ \sum_{t=1}^{T}\sum_{j=1}^{2} \mathbf{1}(t \in \mathbb{T}_i^c \text{ or } j = 2)\frac{U_i(s_{t,i}, \delta)}{s_{t,i}} \right]$$

Now the second part of this sum is easy to deal with as it can be incorporated into the bound of $\mathbf{E}[\text{II}]$ by slightly modifying the bound given by Equation 18 below and changing $2b_l - 1$ to $2b_l + 1$. The rest of the argument remains the same.

### F.2 Lemma F.1

From this section onward we drop the subscript $i$ whenever clear to simplify the notations. In this section we show an upper bound for Term II when there is a value $\underline{p}_i \in (0,1)$ that lower bounds $p_1^i, \cdots, p_T^i$ with probability 1. We then use the restarting trick to extend the proof to the case when $\underline{p}_i$ is random in Theorem 5.2.

**Lemma F.1** (Fixed $\underline{p}_i$). *Let $\underline{p}_i \in (0,1)$ be such that $\frac{1}{\rho_i} = \underline{p}_i \leq p_1^i, \cdots, p_T^i$ with probability one, then,* $\mathbf{E}\left[\mathrm{II}\right] \leq 4\rho_i\, U_i(T/\rho_i, \delta) \log T + \delta T$.

*Proof of Lemma F.1.* Since $\mathbf{E}\left[\mathrm{II}\right] \leq \mathbf{E}\left[\mathbf{1}\{\mathcal{E}\}\mathrm{II}\right] + \delta T$, we focus on bounding $\mathbf{E}\left[\mathbf{1}\{\mathcal{E}\}\mathrm{II}\right]$. since base $i$ is $(U, T, \delta)-$bounded, $\mathbf{E}\left[R_i^{(1)}(\mathbb{T}_i)\mathbf{1}(\mathcal{E})\right] \leq \mathbf{E}\left[U_i(\delta, n_T^i)\mathbf{1}(\mathcal{E})\right]$. We proceed to bound the regret corresponding to the remaining terms in $\mathrm{II}_0$:

$$\mathbf{E}\left[\mathrm{II}_0\mathbf{1}(\mathcal{E})\right] = \mathbf{E}\left[\sum_{l=1}^{n_T^i+1} \mathbf{1}\{\mathcal{E}\}(2b_l - 1)\mathbf{E}\left[r_{t_l,i}^{(2)}|\mathcal{F}_{t_{l-1}}\right]\right]$$

$$\leq \mathbf{E}\left[\sum_{l=1}^{n_T^i+1} \mathbf{1}\{\mathcal{E}\}(2b_l - 1)\frac{U_i(l, \delta/2M)}{l}\right] \tag{18}$$

The multiplier $2b_l - 1$ arises because the policies proposed by the base algorithm during the rounds it is not selected by $\mathcal{M}$ satisfy $\pi_{t,i}^{(1)} = \pi_{t,i}^{(2)} = \pi_{t_l,i}^{(2)}$ for all $l \leq n_i^T + 1$ and $t = t_{l-1} + 1, \cdots, t_l - 1$. The factorization is a result of conditional independence between $\mathbf{E}\left[r_{t_l,i}^{(2)}|\mathcal{F}_{t_{l-1}}\right]$ and $\mathbf{E}\left[b_l|\mathcal{F}_{t_{l-1}}\right]$ where $\mathcal{F}_{t_{l-1}}$ already includes algorithm $\tilde{B}_i$ update right after round $t_{l-1}$. The inequality holds because $\tilde{\mathcal{B}}_i$ is $(U_i, \frac{\delta}{2M}, \mathcal{T}^{(2)})-$smooth and therefore satisfies Equation 3 on event $\mathcal{E}$.

Recall that as a consequence of Equation 17 we have $\mathbf{E}\left[\mathrm{II}\right] \leq \mathbf{E}\left[R_i^{(1)}(\mathbb{T}_i)\mathbf{1}(\mathcal{E}) + \mathrm{II}_0\mathbf{1}\{\mathcal{E}\}\right] + \delta T$. The first term is bounded by $\mathbf{E}\left[U_i(n_T^i, \delta)\mathbf{1}(\mathcal{E})\right]$ while the second term satisfies the bound in (18). Let $u_l = \frac{U_i(l, \delta/2M)}{l}$. By Lemma C.1, $\sum_{l=1}^t u_l \geq U_i(t, \delta/M)$ for all $t$, and so,

$$\mathbf{E}\left[\mathbf{1}\{\mathcal{E}\}U_i(n_T^i, \delta)\right] \leq \mathbf{E}\left[\sum_{l=1}^{n_T^i+1} \mathbf{1}\{\mathcal{E}\}u_l\right]. \tag{19}$$

By (18) and (19),

$$\mathbf{E}\left[R_i^{(1)}(\mathbb{T}_i)\mathbf{1}(\mathcal{E}) + \mathrm{II}_0\mathbf{1}\{\mathcal{E}\}\right] \leq \mathbf{E}\left[\sum_{l=1}^{n_T^i+1} \mathbf{1}\{\mathcal{E}\}2b_l u_l\right].$$

Let $a_l = \mathbf{E}[b_l]$ for all $l$. Consider a master algorithm that uses $\underline{p}_i$ instead of $p_t^i$. In this new process let $t_l'$ be the corresponding rounds when the base is selected, $\bar{n}_T^i$ be the total number of rounds the base is selected, and $c_l = \mathbf{E}\left[t_l' - t_{l-1}'\right]$. Since $\underline{p}_i \leq p_t^i$ for all $t$ it holds that $\sum_{l=1}^j a_l \leq \sum_{l=1}^j c_l$ for all $j$. If we use the same coin flips used to generate $t_l$ to generate $t_l'$, we observe that $t_l' \subset t_l$ and $\bar{n}_T^i \leq n_T^i$. Let $f : \mathbb{R} \to [0,1]$ be a decreasing function such that for integer $i$, $f(i) = u_i$. Then $\sum_{l=1}^{n_T^i+1} a_l u_l$ and $\sum_{l=1}^{\bar{n}_T^i+1} c_l u_l$ are two estimates of integral $\int_0^T f(x)dx$. Given that $t_l' \subset t_l$ and $u_l$ is a decreasing sequence in $l$,

$$\sum_{l=1}^{n_T^i+1} \mathbf{E}\left[t_l - t_{l-1}\right] u_l \leq \sum_{l=1}^{\bar{n}_T^i+1} \mathbf{E}\left[t_l' - t_{l-1}'\right] u_l,$$

and thus

$$\mathbf{E}\left[R_i^{(1)}(\mathbb{T}_i)\mathbf{1}(\mathcal{E}) + \mathrm{II}_0\mathbf{1}\{\mathcal{E}\}\right] \leq \mathbf{E}\sum_{l=1}^{\bar{n}_T^i+1} 2\mathbf{E}\left[t_l' - t_{l-1}'\right] u_l.$$

We proceed to upper bound the right hand side of this inequality:

$$\mathbf{E}\left[\sum_{l=1}^{\bar{n}_T^i+1} u_l \mathbf{E}\left[t_l' - t_{l-1}'\right]\right] \leq \mathbf{E}\left[\sum_{l=1}^{\bar{n}_T^i+1} \frac{u_l}{\underline{p}_i}\right]$$

$$\leq 2\rho_i U_i(T/\rho_i, \delta)\log(T).$$

The first inequality holds because $\mathbf{E}\left[t_l' - t_{l-1}'\right] \leq \frac{1}{\underline{p}_i}$ and the second inequality follows by concavity of $U_i(t, \delta)$ as a function of $t$. The proof follows. $\qquad\square$

### F.3 Proof of Theorem 5.2

We use the restarting trick to extend Lemma F.1 to the case when the lower bound $\underline{p}_i$ is random (more specifically the algorithm (CORRAL) will maintain a lower bound that in the end will satisfy $\underline{p}_i \approx \min_t p_t^i$) in Theorem 5.2. We restate the theorem statement here for convenience.

**Theorem F.2** (Theorem 5.2 ).

$$\mathbf{E}\left[\text{II}\right] \leq \mathcal{O}(\mathbf{E}\left[\rho_i, U_i(T/\rho_i, \delta)\log T\right] + \delta T(\log T + 1)).$$

*Here, the expectation is over the random variable* $\rho_i = \max_t \frac{1}{p_t^i}$. *If* $U(t, \delta) = t^\alpha c(\delta)$ *for some* $\alpha \in [1/2, 1)$ *then,* $\mathbf{E}\left[\text{II}\right] \leq 4\frac{2^{1-\alpha}}{2^{1-\alpha}-1}T^\alpha c(\delta)\mathbf{E}\left[\rho_i^{1-\alpha}\right] + \delta T(\log T + 1).$

**Restarting trick:** Initialize $\underline{p}_i = \frac{1}{2M}$. If $p_t^i < \underline{p}_i$, set $\underline{p}_i = \frac{p_t^i}{2}$ and restart the base.

*Proof of Theorem 5.2.* The proof follows that of Theorem 15 in [2]. Let $\ell_1, \cdots, \ell_{d_i} < T$ be the rounds where Line 10 of the CORRAL is executed. Let $\ell_0 = 0$ and $\ell_{d_i+1} = T$ for notational convenience. Let $e_l = [\ell_{l-1} + 1, \cdots, \ell_l]$. Denote by $\underline{p}_{i,\ell_l}$ the probability lower bound maintained by CORRAL during timesteps $t \in [\ell_{l-1}, \cdots, \ell_l]$ and $\rho_{i,\ell_l} = 1/\underline{p}_{i,\ell_l}$. In the proof of Lemma 13 in [2], the authors prove $d_i \leq \log(T)$ with probability one. Therefore,

$$\mathbf{E}\left[\text{II}\right] = \sum_{l=1}^{\lceil \log(T) \rceil} \mathbb{P}(\underbrace{d_i + 1 \geq l}_{I(l)})\mathbf{E}\left[R_i^{(1)}(e_l) + R_i^{(2)}(e_l)|d_i + 1 \geq l\right]$$

$$\leq \log T \sum_{l=1}^{\lceil \log(T) \rceil} \mathbb{P}(I(l))\mathbf{E}\left[4\rho_{i,\ell_l}U_i(T/\rho_{i,\ell_l}, \delta)|I(l)\right] + \delta T(\log T + 1)$$

$$= \log T \mathbf{E}\left[\sum_{l=1}^{b_i+1} 4\rho_{i,\ell_l}U_i(T/\rho_{i,\ell_l}, \delta)\right] + \delta T(\log T + 1).$$

The inequality is a consequence of Lemma F.1 applied to the restarted segment $[\ell_{l-1}, \cdots, \ell_l]$. This step is valid because by assumption $\frac{1}{\rho_{i,\ell_l}} \leq \min_{t \in [\ell_{l-1}, \cdots, \ell_l]} p_t$.

If $U_i(t, \delta) = t^\alpha c(\delta)$ for some function $c : \mathbb{R} \to \mathbb{R}^+$, then $\rho_i U(T/\rho_i, \delta) = \rho_i^{1-\alpha}T^\alpha c(\delta)$. And therefore:

$$\mathbf{E}\left[\sum_{l=1}^{b_i+1} \rho_{i,\ell_l}U_i(T/\rho_{i,\ell_l}, \delta)\right] \leq T^\alpha g(\delta)\mathbf{E}\left[\sum_{l=1}^{b_i+1} \rho_{i,\ell_l}^{1-\alpha}\right]$$

$$\leq \frac{2^{\bar{\alpha}}}{2^{\bar{\alpha}} - 1}T^\alpha c(\delta)\mathbf{E}\left[\rho_i^{1-\alpha}\right]$$

Where $\bar{\alpha} = 1 - \alpha$. The last inequality follows from the same argument as in Theorem 15 in [2]. □

## G   Total Regret

*Proof of Theorem 5.3.* For the CORRAL master,

$$\mathbf{E}\left[\text{I}\right] \leq \mathbf{E}\left[\text{I}_A\right] + \mathbf{E}\left[\text{I}_B\right] \leq O\left(\frac{M \ln T}{\eta} + T\eta\right) - \frac{\mathbf{E}\left[\rho\right]}{40\eta \ln T} + 8\sqrt{MT \log(\frac{4TM}{\delta})}$$

Using Theorem 5.2 to control term II, the total regret of CORRAL is:

$$R(T) \leq O\left(\frac{M \ln T}{\eta} + T\eta\right) - \mathbf{E}\left[\frac{\rho}{40\eta \ln T} - 2\rho\, U(T/\rho, \delta) \log T\right] + \delta T + 8\sqrt{MT \log(\frac{4TM}{\delta})}$$

$$\leq O\left(\frac{M \ln T}{\eta} + T\eta\right) - \mathbf{E}\left[\frac{\rho}{40\eta \ln T} - 2\rho^{1-\alpha}T^\alpha c(\delta) \log T\right] + \delta T + 8\sqrt{MT \log(\frac{4TM}{\delta})}$$

$$\leq \tilde{O}\left(\sqrt{MT} + \frac{M}{\eta} + T\eta + Tc(\delta)^{\frac{1}{\alpha}}\eta^{\frac{1-\alpha}{\alpha}}\right) + \delta T,$$

where the last step is by maximizing the function over $\rho$. Choose $\delta = 1/T$. When both $\alpha$ and $c(\delta)$ are known, choose $\eta = \frac{M^\alpha}{c(\delta)T^\alpha}$. When only $\alpha$ is known, choose $\eta = \frac{M^\alpha}{T^\alpha}$.

For the EXP3.P master, if $p \leq \frac{1}{2k}$:

$$\mathbf{E}\left[I\right] \leq \mathbf{E}\left[I_A\right] + \mathbf{E}\left[I_B\right] \leq \tilde{O}\left(MTp + \frac{\log(k\delta^{-1})}{p} + \sqrt{MT\log(\frac{4TM}{\delta})}\right)$$

Using Lemma F.1 to control term II, we have the total regret of EXP3.P when $\delta = 1/T$:

$$R(T) = \tilde{O}(\sqrt{MT} + MTp + \frac{1}{p} + \frac{1}{p}U_i(Tp, \delta)) \ .$$
$$= \tilde{O}(\sqrt{MT} + MTp + T^\alpha p^{\alpha-1}c(\delta))$$

When both $\alpha$ and $c(\delta)$ are known, choose $p = T^{-\frac{1-\alpha}{2-\alpha}}M^{-\frac{1}{2-\alpha}}c(\delta)^{\frac{1}{2-\alpha}}$. When only $\alpha$ is known, choose $p = T^{-\frac{1-\alpha}{2-\alpha}}M^{-\frac{1}{2-\alpha}}$. We then have the following regret:

| | EXP3.P | CORRAL |
|---|---|---|
| General | $\tilde{O}\left(\sqrt{MT} + MTp + T^\alpha p^{\alpha-1}c(\delta)\right)$ | $\tilde{O}\left(\sqrt{MT} + \frac{M}{\eta} + T\eta + T\,c(\delta)^{\frac{1}{\alpha}}\eta^{\frac{1-\alpha}{\alpha}}\right)$ |
| Known $\alpha$ Known $c(\delta)$ | $\tilde{O}\left(\sqrt{MT} + M^{\frac{1-\alpha}{2-\alpha}}T^{\frac{1}{2-\alpha}}c(\delta)^{\frac{1}{2-\alpha}}\right)$ | $\tilde{O}\left(\sqrt{MT} + M^\alpha T^{1-\alpha} + M^{1-\alpha}T^\alpha c(\delta)\right)$ |
| Known $\alpha$ Unknown $c(\delta)$ | $\tilde{O}\left(\sqrt{MT} + M^{\frac{1-\alpha}{2-\alpha}}T^{\frac{1}{2-\alpha}}c(\delta)\right)$ | $\tilde{O}\left(\sqrt{MT} + M^\alpha T^{1-\alpha} + M^{1-\alpha}T^\alpha c(\delta)^{\frac{1}{\alpha}}\right)$ |

$\square$

# H    Lower Bounds (Proofs of Section 6)

Here we state the proofs of Section 6.

*Proof of Theorem 6.1.* Consider a stochastic 2-arm bandit problem where the best arm has expected reward $1/2$ and the second best arm has expected reward $1/4$. We construct base algorithms $\mathcal{B}_1, \mathcal{B}_2$ as follows. $\mathcal{B}_1$ always chooses the optimal arm and its expected instantaneous reward is $1/2$. $\mathcal{B}_2$ chooses the second best arm at time step $t$ with probability $\frac{4c}{\sqrt{t+2}\log(t+2)}$ ($c$ will be specified later), and chooses the best arm otherwise. The expected reward at time step $t$ of $\mathcal{B}_2$ is $\frac{1}{2} - \frac{c}{\sqrt{t+2}\log(t+2)}$.

Let $A^*$ be uniformly sampled from $\{1, 2\}$. Consider two environments $\nu_1$ and $\nu_2$ for the master, each made up of two base algorithms $\tilde{\mathcal{B}}_1, \tilde{\mathcal{B}}_2$. Under $\nu_1$, $\tilde{\mathcal{B}}_1$ and $\tilde{\mathcal{B}}_2$ are both instantiations of $\mathcal{B}_1$. Under $\nu_2$, $\tilde{\mathcal{B}}_{A^*}$, where $A^*$ is a uniformly sampled index in $\{1, 2\}$, is a copy of $\mathcal{B}_1$ and $\tilde{\mathcal{B}}_{3-A^*}$ is a copy of $\mathcal{B}_2$.

Let $\mathbb{P}_1, \mathbb{P}_2$ denote the probability measures induced by interaction of the master with $\nu_1$ and $\nu_2$ respectively. Let $\tilde{\mathcal{B}}_{A_t}$ denote the base algorithm chosen by the master at time $t$. We have $\mathbb{P}_1(A_t \neq A^*) = \frac{1}{2}$ for all $t$, since the learner has no information available to identify which algorithm is considered optimal. By Pinskers' inequality we have

$$\mathbb{P}_2(A_t \neq A^*) \geq \mathbb{P}_1(A_t \neq A^*) - \sqrt{\frac{1}{2}\text{KL}(\mathbb{P}_1 || \mathbb{P}_2)}$$

By the divergence decomposition [see 12, proof of Lemma 15.1 for the decomposition technique] and using that for $\Delta < \frac{1}{4}$ : $\text{KL}(\frac{1}{2}, \frac{1}{2} - \Delta) \leq 3\Delta^2$ (Lemma C.3), we have

$$\text{KL}(\mathbb{P}_1 || \mathbb{P}_2) = \sum_{t=2}^{\infty} \frac{1}{2}\text{KL}\left(\frac{1}{2}, \frac{1}{2} - \frac{c}{\sqrt{t+1}\log(t+1)}\right)$$
$$\leq \sum_{t=2}^{\infty} \frac{3c^2}{2t\log(t)^2} \leq 3c^2 \ .$$

Picking $c = \sqrt{\frac{1}{24}}$ leads to $\mathbb{P}_2(A_t \neq A^*) \geq \frac{1}{4}$, and the regret in environment $\nu_2$ is lower bounded by

$$R(T) \geq \sum_{t=1}^{T} \mathbb{P}_2(A_t \neq A^*) \frac{c}{\sqrt{t+1}\log(t+1)}$$

$$\geq \frac{c}{4\log(T+1)} \sum_{t=1}^{T} \frac{1}{\sqrt{t+1}} = \Omega(\frac{\sqrt{T}}{\log(T)}).$$

$\square$

*Proof.* Let the set of arms be $\{a_1, a_2, a_3\}$. Let $x$ and $y$ be such that $0 < x < y \leq 1$. Let $\Delta = T^{x-1+(y-x)/2}$. Define two environment $\mathcal{E}_1$ and $\mathcal{E}_2$ with reward vectors $\{1, 1, 0\}$ and $\{1 + \Delta, 1, 0\}$ for $\{a_1, a_2, a_3\}$, respectively. Let $B_1$ and $B_2$ be two base algorithms defined by the following fixed policies when running alone in $\mathcal{E}_1$ or $\mathcal{E}_2$:

$$\pi_1 = \begin{cases} a_2 & \text{w.p. } 1 - T^{x-1} \\ a_3 & \text{w.p. } T^{x-1} \end{cases} \quad , \quad \pi_2 = \begin{cases} a_2 & \text{w.p. } 1 - T^{y-1} \\ a_3 & \text{w.p. } T^{y-1} \end{cases} .$$

We also construct base $B_2'$ defined as follows. Let $c_2 > 0$ and $\epsilon_2 = (y-x)/4$ be two constants. Base $B_2'$ mimics base $B_2$ when $t \leq c_2 T^{x-y+1+\epsilon_2}$, and picks arm $a_1$ when $t > c_2 T^{x-y+1+\epsilon_2}$. The instantaneous rewards of $B_1$ and $B_2$ when running alone are $r_t^1 = 1 - T^{x-1}$ and $r_t^2 = 1 - T^{y-1}$ for all $1 \leq t \leq T$. Next, consider model selection with base algorithms $B_1$ and $B_2$ in $\mathcal{E}_1$. Let $T_1$ and $T_2$ be the number of rounds that $B_1$ and $B_2$ are chosen, respectively.

First, assume case (1): There exist constants $c > 0$, $\epsilon > 0$, $p \in (0,1)$, and $T_0 > 0$ such that with probability at least $p$, $T_2 \geq cT^{x-y+1+\epsilon}$ for all $T > T_0$.

The regret of base $B_1$ when running alone for $T$ rounds is $T \cdot T^{x-1} = T^x$. The regret of the model selection method is at least

$$p \cdot T_2 \cdot T^{y-1} \geq p \cdot cT^{x-y+1+\epsilon} \cdot T^{y-1} = p \cdot c \cdot T^{x+\epsilon} .$$

Given that the inequality holds for any $T > T_0$, it proves the statement of the lemma in case (1).

Next, we assume the complement of case (1): For all constants $c > 0$, $\epsilon > 0$, $p \in (0,1)$, and $T_0 > 0$, with probability at least $p$, $T_2 < cT^{x-y+1+\epsilon}$ for some $T > T_0$.

Let $T$ be any such time horizon. Consider model selection with base algorithms $B_1$ and $B_2'$ in environment $\mathcal{E}_2$ for $T$ rounds. Let $T_1'$ and $T_2'$ be the number of rounds that $B_1$ and $B_2'$ are chosen. Note that $B_2$ and $B_2'$ behave the same for $c_2 T^{x-y+1+\epsilon}$ time steps, and that $B_1$ and $B_2$ never choose action $a_1$. Therefore for the first $c_2 T^{x-y+1+\epsilon_2}$ time steps, the model selection strategy that selects between $B_1$ and $B_2'$ in $\mathcal{E}_2$ behaves the same as when it runs $B_1$ and $B_2$ in $\mathcal{E}_1$. Therefore with probability $p > 1/2$, $T_2' < c_2 T^{x-y+1+\epsilon_2}$, which implies $T_1' > T/2$.

In environment $\mathcal{E}_2$, the regret of base $B_2'$ when running alone for $T$ rounds is bounded as

$$(\Delta + T^{y-1})c_2 T^{x-y+1+\frac{y-x}{4}} = c_2 T^{\frac{5x-y}{4}} + c_2 T^{\frac{3x+y}{4}} < 2c_2 T^{\frac{3x+y}{4}}$$

Given that with probability $p > 1/2$, $T_1' > T/2$, the regret of the learner is lower bounded as,

$$p(\Delta + T^{x-1}) \cdot \frac{T}{2} > \frac{1}{2}(T^{x-1+\frac{y-x}{2}} + T^{x-1}) \cdot \frac{T}{2} < \frac{1}{2} T^{\frac{x+y}{2}} ,$$

which is larger than the regret of $B_2'$ running alone because $\frac{3x+y}{4} < \frac{x+y}{2}$. The statement of the lemma follows given that for any $T_0$ there exists $T > T_0$ so that the model selection fails. $\square$

# I   Applications (Proofs of Section 4)

Here we state the proofs of Section 4.

## I.1 Misspecified Contextual Linear Bandit

*Proof of Theorem 4.1.* From Lemma D.2, for UCB, $U(T, \delta) = O(\sqrt{Tk} \log \frac{Tk}{\delta})$. Therefore from Theorem 5.3, running CORRAL with smooth UCB results in the following regret bound:

$$\tilde{O}\left(\sqrt{MT} + \frac{M \ln T}{\eta} + T\eta + T\left(\sqrt{k} \log \frac{Tk}{\delta}\right)^2 \eta\right) + \delta T.$$

If we choose $\delta = 1/T$ and hide some log factors, we get $\tilde{O}\left(\sqrt{T} + \frac{1}{\eta} + Tk\eta\right)$.

For the modified LinUCB bases in [13] or [20] or the G-optimal algorithm [13], $U(t, \delta) = O(d\sqrt{t} \log(1/\delta) + \epsilon\sqrt{dt})$. From the proof of Theorem 5.3 and substitute $\delta = 1/T$:

$$R(T) \leq O\left(\sqrt{MT \log(\frac{4TM}{\delta})} + \frac{M \ln T}{\eta} + T\eta\right) - \mathbf{E}\left[\frac{\rho}{40\eta \ln T} - 2\rho \, U(T/\rho, \delta) \log T\right] + \delta T$$

$$\leq \tilde{O}\left(\sqrt{MT} + \frac{M \ln T}{\eta} + T\eta\right) - \mathbf{E}\left[\frac{\rho}{40\eta \ln T} - 2\rho \, (d\sqrt{\frac{T}{\rho}} \log(1/\delta) + \epsilon\sqrt{d}\frac{T}{\rho}) \log T\right]$$

$$\leq \tilde{O}\left(\sqrt{MT} + \frac{M \ln T}{\eta} + T\eta\right) - \mathbf{E}\left[\frac{\rho}{40\eta \ln T} - 2d\sqrt{T\rho} \log(1/\delta) \log T\right] + 2\epsilon\sqrt{dT} \log T$$

Maximizing over $\rho$, and running CORRAL with smooth modified LinUCB results in $\tilde{O}\left(\sqrt{T} + \frac{1}{\eta} + Td^2\eta + \epsilon\sqrt{dT}\right)$ regret bound.

For the misspecified linear bandit problem, we use $M = O(\log(T))$ LinUCB base with $\epsilon$ defined in the grid, and choose $\eta = \frac{1}{\sqrt{T}d}$. The resulting regret will be $\tilde{O}\left(\sqrt{T}d + \epsilon\sqrt{dT}\right)$.

When the action sets are fixed, by the choice of $\eta = \frac{1}{\sqrt{T}d}$, the regret of CORRAL with 1 smooth UCB and 1 G-optimal base will be:

$$\tilde{O}\left(\min\left\{\sqrt{T}\left(d + \frac{k}{d}\right), \sqrt{T}d + \epsilon\sqrt{dT}\right\}\right).$$

If $\sqrt{k} > d$, the above expression becomes $\tilde{O}\left(\min\left(\sqrt{T}\frac{k}{d}, \sqrt{T}d + \epsilon\sqrt{dT}\right)\right)$ □

## I.2 Contextual Bandits with Unknown Dimension

**Linear Contextual Bandit.** First we consider the linear contextual bandit problem with unknown dimension $d_*$. From Lemma D.3 and Lemma D.4, for linear contextual bandit, LinUCB is $(U, \delta, T)$-bounded with $U(t, \delta) = O(d\sqrt{t} \log(1/\delta))$ for infinite action sets and $U(t, \delta) = O(\sqrt{dt} \log^3(kT \log(T)/\delta))$ for finite action sets. Choose $\delta = 1/T$ and ignore the log factor, $U(t, \delta) = \tilde{O}(d\sqrt{t})$ for infinite action sets and $U(t, \delta) = \tilde{O}(\sqrt{dt})$ for finite action sets.

Then $U(t) = c(\delta)t^\alpha$ with $\alpha = 1/2$ and $c(\delta) = \tilde{O}(d)$ for infinite action sets, and $c(\delta) = \tilde{O}(\sqrt{d})$ for finite action sets. When $d_*$ is unknown, a direct application of Theorem 5.3 will yield the following regrets:

| | Linear contextual bandit | |
| --- | --- | --- |
| | Unknown $d_*$ | |
| | Finite action sets | Infinite action sets |
| Foster et al. [8] | $\tilde{O}(T^{2/3}k^{1/3}d_*^{1/3})$ or $\tilde{O}(k^{1/4}T^{3/4} + \sqrt{kTd_*})$ | N/A |
| EXP3.P | $\tilde{O}(d_*^{\frac{1}{2}}T^{\frac{2}{3}})$ | $\tilde{O}(d_*T^{\frac{2}{3}})$ |
| CORRAL | $\tilde{O}\left(d_*\sqrt{T}\right)$ | $\tilde{O}\left(d_*^2\sqrt{T}\right)$ |

Now consider the misspecified linear contextual bandit problem with unknown $d_*$ and $\epsilon_*$. We use the modified LinUCB bases [13, 20]. Using the calculation in the proof of Theorem 4.1 in Section I.1, using CORRAL with a smooth modified LinUCB base with parameters $(d, \epsilon)$ in the grids results in $\tilde{O}\left(\frac{1}{\eta} + Td^2\eta + \epsilon\sqrt{d}T\right)$ regret. Since $d$ is unknown, choosing $\eta = 1/\sqrt{T}$ yields the regret $\tilde{O}\left(\sqrt{T}d_*^2 + \epsilon\sqrt{d}T\right)$.

Using EXP3.P with a smooth modified LinUCB base with parameters $(d, \epsilon)$ in the grids results in:

$$R(T) = \tilde{O}\left(\sqrt{MT} + MTp + \frac{1}{p} + \frac{1}{p}U_i(Tp, \delta)\right) .$$
$$= \tilde{O}\left(\sqrt{MT} + MTp + \frac{1}{p} + \frac{1}{p}\left(d\sqrt{Tp} + \epsilon\sqrt{d}Tp\right)\right) .$$
$$= \tilde{O}\left(\sqrt{MT} + MTp + \frac{d\sqrt{T}}{p} + \epsilon\sqrt{d}T\right) .$$

Since $d_*$ is unknown, choosing $p = T^{-1/3}$ yields the regret bound $\tilde{O}(T^{\frac{2}{3}}d_* + \epsilon_*\sqrt{d}T)$.

| | Misspecified linear contextual bandit |
| --- | --- |
| | Unknown $d_*$ and $\epsilon_*$ |
| Foster et al. [8] | N/A |
| EXP3.P | $\tilde{O}(T^{\frac{2}{3}}d_* + \epsilon_*\sqrt{d}T)$ |
| CORRAL | $\tilde{O}\left(\sqrt{T}d_*^2 + \epsilon_*\sqrt{d}T\right)$ |

**Nonparametric Contextual Bandit.** When the context dimension $n_*$ is known, [9] present an algorithm with $U(T, \delta) = \tilde{O}\left(T^{\frac{1+n_*}{2+n_*}}\right)$ when $\delta = 1/T$. We use this algorithm with value of $n$ in the grid $[b^0, b^1, b^2, ..., b^{\log_b(N)}]$ for some $b > 1$ and applying Theorem 5.3 with $p = T^{-1/3}$ for EXP3.P and $\eta = T^{-1/2}$. Let $n_0$ be the value in the grid such that $n_* \in [n_0/b, n_0]$. Then $n_0 \leq bn_*$. We will have regret $\tilde{O}\left(T^{\frac{1+n_0}{2+n_0} + \frac{1}{3(2+n_0)}}\right)$ for EXP3.P and $\tilde{O}\left(T^{\frac{1+2n_0}{2+2n_0}}\right)$ for CORRAL since $n_0$ is the minimum value in the grid that will have the regret bound in [9]. Since $n_0 \leq bn_*$ the regret is upper bounded by $\tilde{O}\left(T^{\frac{1+bn_*}{2+bn_*} + \frac{1}{3(2+bn_*)}}\right)$ for EXP3.P and $\tilde{O}\left(T^{\frac{1+2bn_*}{2+2bn_*}}\right)$ for CORRAL.

Alternatively, we can also use $N$ base algorithms with each value of $n \in [N]$. Since $n_*$ will be contained in one of the base algorithm, the regret achieved by EXP3.P with $p = N^{-1/2}T^{-1/3}$ and CORRAL with $\eta = (NT)^{-1/2}$ will be $\tilde{O}\left(\sqrt{N}T^{2/3} + N^{\frac{1}{2+n_*}}T^{\frac{1+n_*}{2+n_*} + \frac{1}{3(2+n_*)}}\right)$ for EXP3.P and $\tilde{O}\left(\sqrt{NT} + N^{\frac{1}{4+2n_*}}T^{\frac{1+2n_*}{2+2n_*}}\right)$ for CORRAL.

### I.3 Tuning the Exploration Rate of $\epsilon$-greedy

*Proof of Theorem 4.3.* From Lemma D.5, we lower bound the smallest gap by $1/T$ (because the gaps smaller than $1/T$ will cause constant regret in $T$ time steps) and choose $\delta = 1/T^5$. From Theorem 5.3, the regret is $\tilde{O}(T^{2/3})$ when $k > 2$ and $\tilde{O}(T^{1/2})$ when $k = 2$ with the base running alone.

Next we show that the best value of $c$ in the exponential grid gives a regret that is within a constant factor of the regret above where we known the smallest non-zero gap $\Delta_*$. An exploration rates can be at most $kT$. Since $\frac{5K}{\Delta_*^2} > 1$, we need to search only in the interval $[1, KT]$. Let $c_1$ be the element in the exponential grid such that $c_1 \leq c^* \leq 2c_1$. Then $2c_1 = \gamma c^*$ where $\gamma < 2$ is a constant, and therefore using $2c_1 = \gamma c^*$ will give a regret up to a constant factor of the optimal regret. $\square$

### I.4 Reinforcement Learning

*Proof of Theorem 4.4.* [11] obtain the high probability bound $\tilde{O}(\sqrt{d^3 H^3 T})$ for LSVI-UCB where $H$ is the length of each episode. Recall that we focus on the episodic setting. We treat each episode

and the re-sampling of a policy deployed by the algorithm in previous episodes as a single unit. The result then follows from Theorem 5.3 by setting the CORRAL learning rate as $\eta = \frac{M^{1/2}}{T^{1/2}d^{3/2}H^{3/2}}$. $\quad\square$

## Footnotes

[3]This choice of $c$ is robust to multiplication by a constant.