[Reviews · NeurIPS 2020]

Review 1

Summary and Contributions: This paper studies the problem of algorithm selection in stochastic contextual bandits. The two master algorithms, the CORRAL and the EXP3.P algorithms, need the based algorithms satisfy certain stability condition. To solve this issue, a smoothed algorithm is proposed for transforming any contextual bandit algorithms to be compatible with the two masters. Then the proposed model selection algorithm is applied to several model selection problems in bandit and reinforcement learning. Finally, two kinds of lower bounds are proposed for proving the optimality of the proposed algorithm. The main contributions of this paper include a smoothed algorithm, several interesting applications, and some solid theoretical results.

Strengths: The problem is significant to the bandit learning community. The algorithmic contributions and the theoretical contributions are original and solid.

Weaknesses: The experimental results seems insufficiency. The experimental results in Figure 2 show that the proposed algorithm with the exponential grid technique does not obtain superior performance. There still exists a large gap from the best base algorithm.

Correctness: All of the claims and the experimental results in the main text are sound.

Clarity: As a whole, this paper is well written, except for many undefined notations in problem statement. The main results are presented clearly. Besides, there is a typos. line 145: Experiment (Figure 4) ===> Experiment (Figure 1)

Relation to Prior Work: This paper discusses the related work in detail. It is clear that the contributions are different from that of the prior work.

Reproducibility: Yes

Additional Feedback: One of key techniques adapted by the proposed model selection algorithm is to divide the range of unknown parameter into an exponential grid. Although the technique is not a new idea, it induces improved theoretical results. However, I have a concern about this technique. The algorithm seems not adaptive, since it needs to know the range of unknown parameter. The selection of the range may be critical to the practical performance of the model selection algorithm. The authors may provide some comments on the selection of the range, and show some empirical results. [After rebuttal] Thank you for your response. The feedback erases my concern. Thus I decide to improve my score.


Review 2

Summary and Contributions: [after rebuttal] I read the other reviews and the authors rebuttal. I still consider this work well done and relevant for the community. === The authors investigate the bandit model selection problem and propose and alternative solution compared to state of the art approaches. Considering the stochastic domain they also show a lower bound to the regret of the master algorithm which is independent of the base algorithms bounds.

Strengths: The considered problem started receiving a lot of attention during the last years. In this work the authors proposed a novel approach to the contextual bandit model selection problem. The main contribution is given by the introduction of a smoothing transformation wrapping the base algorithms which does not require to modify the base learners as it was necessary in the state of the art work.

Weaknesses: I did not found clear weaknesses. I found the contributions to be relevant for the bandit community in particular given the recent interest in the model selection problem.

Correctness: I did not check all the proofs in the supplementary material. The sketches of the proof presented in the main paper seems reasonable. The experimental evaluation has been carried out properly.

Clarity: The paper is well written, sections are self-explained and the notation is clear. The settings considered for the experiments are well reported and the obtained results are clear.

Relation to Prior Work: The authors make a clear comparison of the proposed approach with respect to previously existing methods. I have just a small comment, checking the references I found this work "Weighted linear bandits for non-stationary environments." regarding non-statinoary bandits but I was not able to find its citation and its connection.

Reproducibility: Yes

Additional Feedback:


Review 3

Summary and Contributions: The paper tackles the problem of sequentially choosing among a set of "base" algorithms (each appropriate for different type of environment) in order to minimise regret over an unknown environment - i.e. discover the best "base" algorithm for an unknown setting while minimising regret. The paper proposes a novel smoothing step that is able to allow the existing CORRAL algorithm to integrate a wider range of base algorithms and provide regret bounds improving on the state-of-the-art in some specific settings - the misspecified contextual linear bandit, contextual linear bandits of unknown dimensions for instance. Other settings are addressed as well but the guarantees there are lacking context on the performance limits in the state-of-the-art.

Strengths: The paper addresses an interesting problem and proposes a relatively simple adaptation to existing methods that yield improvements in the existing theoretical results in several interesting settings. I think the results are novel and significant.

Weaknesses: I believe the paper could present more context around the results in sections 4.2 through 4.4. What other results exist in these settings? Similarly, the numerical experiments are not discussed at all and are hard to interpret. I find it hard to draw conclusions from these experiments. I would recommend the description of the algorithm be moved ahead of section 4. Section 4 is hard to judge on the first read as the algorithm generating the claimed results is not yet presented. The assumption that Base algorithms only have access to rewards of rounds when they are selected, seems artificial and restrictive. It would be helpful to gauge the applicability of the setting if there were a few examples of real life scenarios where this assumption is satisfied. Edit after rebuttal: I'd like to thank the authors for their feedback. I found the response eased most of my concerns and clarified a series of misunderstandings. I will therefore raise my score to 7 (accept). In detail: - Thank you for pointing out the implementation details in the Appendix. I missed them the first time around. - I still find the experiments would benefit from further discussion - thank you for clarifying that using the observations only for the algorithm that was active is a design choice - my concerns regarding correctness below are also eased

Correctness: The empirical methodology is not presented in detail and I think it's not trivial to reproduce the results in all the figures presented here. I have not checked the proofs in the appendix. The claim on line 227-229 needs to be detailed: "Since the instantaneous regret of Step 2 is 1/s times the cumulative regret of Step 1, the cumulative regret of Step 2 over S states is bounded roughly by log(S) * regret of step 1." I find this questionable since if all the samples happened after S rounds, this would be true. But the regret of the base algorithm is sublinear, meaning in earlier rounds, we have higher regret. Most samples are drawn earlier so worse plays are going to be overrepresented in the cumulative regret, whereas only last samples will roughly be bounded as advertised. Is this concern founded or am I missing something? I would like this aspect to be explained more clearly.

Clarity: The paper is generally well written but I would prefer the structure to be revisited and the algorithm presented ahead of the Applications (swap sections 5 and 4).

Relation to Prior Work: I think this can be improved particularly for the results in section 4. I would like to see references to the existing regret bounds in the settings described.

Reproducibility: No

Additional Feedback: I like the problem tackled here and the relative simplicity of the method, though I think the discussion of the results is lacking. More discussion is needed around the context of the theoretical and numerical results in section 4, what are the improvements over the state of the art that the method enables. I would like to be clear where the results improve upon the state of the art and where existing results are recovered (through this more general method). Baselines of other competing algorithms would also be useful to gauge the quality of the numerical experiments. As is, the paper appears too dense. Other questions and remarks: On line 94: what is $i$? In Step 2 of the smoothing procedure, how are cases where the action set changes handled in the sampling from history. Take a linear contextual bandit for example, at every round, the feature vectors change so at round t, we might see completely new arms relative to the history 1...t. How does Step 2 of the algorithm adapt to this? What do lines 6 through 11 in the pseudocode do? It appears the algorithms is selecting arms when it is not chosen by the meta-policy, why is the purpose of this step given the algorithm receives no feedback?

[Author Response · NeurIPS 2020]

We would like to thank the reviewers for their time. We humbly request for more of your time to reevaluate scores, given our responses below. We propose a general procedure to modify bandit algorithms for stochastic contextual bandit problems to be compatible with multiple adversarial bandit algorithms which allows us to obtain previously unattainable model selection regret guarantees that can be applied to a wide variety of problems in the setting of i.i.d. contexts. These assumptions (stochastic environments and i.i.d. contexts) are fairly standard in the literature. Because of its portability, this allows us to easily plug in multiple existing bandit algorithms and obtain regret guarantees for many problems for which no model selection guarantees were known before. Notably, we are able to obtain meaningful model selection guarantees even when the best base algorithm's regret is not fully known. We summarize our contributions:

**(Section 4.1) Mis-specified contextual bandit with unknown error:** We provide the first solution for the case of changing action sets. For fixed action sets, we improve upon the best existing result (Lattimore et al, 2020), and match the lower bound. To understand this result, consider $k$ arms of dimension $d < \sqrt{k}$. If the arms are linear, then UCB has regret $\tilde{O}(\sqrt{kT})$ and LinUCB has $\tilde{O}(d\sqrt{T})$, so the best base's regret is $\tilde{O}(d\sqrt{T})$. If the arms are not linear, then UCB has regret $\tilde{O}(\sqrt{kT})$ and LinUCB has linear regret, so the best base's regret is $\tilde{O}(\sqrt{kT})$. It is impossible to know the best base's regret without knowing the environment, but our method achieves a regret matching the lower bound.

**(Section 4.2) Linear contextual bandit with unknown dimension:** For finite action sets, we provide a regret bound that does not depend on the number of actions, in contrast to the best existing result (Foster et al, 2019). We provide the first solution for infinite action sets. We provide the first solution when both the dimension and the mis-specified error are unknown. This is also the first result in literature that can combine multiple types of model selection.

**(Section 4.2) Non-parametric contextual bandit with unknown dimension:** we provide the first solution.

**(Section 4.3) Tuning the exploration rate of $\epsilon$-greedy:** we provide the first solution for this problem.

**(Section 4.4) Choosing between multiple feature maps for RL:** we provide the first solution.

**(Appendix A1) Generalized linear bandits with unknown link function:** we provide this problem's first solution.

**(Appendix A2) Bandit with heavy tail:** we provide the first solution for this problem.

**Reviewer 1:** "The selection of the range": The regret is multiplied by at most a factor of the number of bases $M$, which is $\log(B)$ if the upper bound of the parameter is $B$. Therefore $B$ can be chosen quite loosely as long as $\log(B)$ is not too large. In the paper we choose the largest $\epsilon$ to be $100,000$ but in practice such a large $\epsilon$ is unreasonable. Still, even with such a loosely chosen $B$, the performance is reasonably well.

We would like to emphasize that the exponential grid division of the parameter space is not central to our key contributions. How to define the best range and search efficiently in the parameter space is an interesting but separate research topic not in the scope of this paper.

**Reviewer 2:** Thank you for your comments. The citation to the non-stationary bandit paper was left from an earlier version of this submission, and will be removed.

**Reviewer 3:** *Weaknesses:* A) Relation to prior work in Section 4: In Section 4, we show that the strategy can be seamlessly used to solve a number of challenging open problems (reiterated above). We provide the first solution for these problems, and there are no existing solutions to compete with.

B) We would present the full description of the algorithm before Section 4.

C) "The assumption that Base algorithms only have access to rewards of rounds when they are selected...": This is not an assumption but a standard algorithmic design choice (modularity). It is not restrictive because we are able to produce a variety of state-of-the-art results (listed at the beginning). It is not clear if letting the base algorithms share the rewards will increase the performance.

*Correctness:* A) "I think it's not trivial to reproduce the results...": We would like more explanations as to why it would be difficult to reproduce the results. We believe we have provided all the details for our experiments. We have listed all configurations, parameter choices and number of repetitions. We also reproduced the master algorithms in Appendix B for convenience.

B) "...the instantaneous regret of Step 2 is 1/s times...": Let the cumulative regret of step 1 at round $s$ be $U(s)$, then the cumulative regret of step 2 at round $S$ is: $\frac{1}{1}U(1) + \frac{1}{2}U(2) + ... + \frac{1}{S}U(S) < (\frac{1}{1} + \frac{1}{2} + ... + \frac{1}{S})U(S) = \log(S)U(S)$. More details are in the proof in Lemma F1, page 24.

*Additional feedback:* A) "On line 94: what is $i$?": $i$ is the index of the base.

B) "...cases where the action set changes...": In Step 2 of the smoothing procedure, we repeat the policy (which is the way to compute the chosen action), not the action. Even if the action set changes, the policy stays the same.

C) "...lines 6 through 11 in the pseudocode...": this is for the analysis of term II, as explained from line 90-98.

[Meta-Review · NeurIPS 2020]

Three referees reviewed the paper, and initially raised several concerns. However, the rebuttal did address and overcome the reviewer's objections, leading to a unanimous final decision to accept the paper.